# CAKE: CAUSAL AND COLLABORATIVE PROXY-TASKS LEARNING FOR SEMI-SUPERVISED DOMAIN ADAPTATION

## ABSTRACT

Semi-supervised domain adaptation (SSDA) adapts a learner to a new domain by effectively utilizing source domain data and a few labeled target samples. It is a practical yet under-investigated research topic. In this paper, we analyze the SSDA problem from two perspectives that have previously been overlooked, and correspondingly decompose it into two *key subproblems*: *robust domain adaptation (DA) learning* and *maximal cross-domain data utilization*. **(i)** From a causal theoretical view, a robust DA model should distinguish the invariant "concept" (key clue to image label) from the nuisance of confounding factors across domains. To achieve this goal, we propose to generate *concept-invariant samples* to enable the model to classify the samples through causal intervention, yielding improved generalization guarantees; **(ii)** Based on the robust DA theory, we aim to exploit the maximal utilization of rich source domain data and a few labeled target samples to boost SSDA further. Consequently, we propose a collaboratively debiasing learning framework that utilizes two complementary semi-supervised learning (SSL) classifiers to mutually exchange their unbiased knowledge, which helps unleash the potential of source and target domain training data, thereby producing more convincing pseudo-labels. Such obtained labels facilitate cross-domain feature alignment and duly improve the invariant concept learning. In our experimental study, we show that the proposed model significantly outperforms SOTA methods in terms of effectiveness and generalisability on SSDA datasets.

## 1 INTRODUCTION

Domain Adaptation (DA) aims to transfer training knowledge to the new domain (*target $\mathcal{D} = \mathcal{D}_{\mathcal{T}}$*) using the labeled data available from the original domain (*source $\mathcal{D} = \mathcal{D}_{\mathcal{S}}$*), which can alleviate the poor generalization of learned deep neural networks when the data distribution significantly deviates from the original domain Wang & Deng (2018); You et al. (2019); Tzeng et al. (2017). In the DA community, recent works Saito et al. (2019) have shown that the presence of few labeled data from the target domain can significantly boost the performance of deep learning-based models. This observation led to the formulation of Semi-Supervised Domain Adaptation (SSDA), which is a variant of Unsupervised Domain Adaptation (UDA) Venkateswara et al. (2017) to facilitate model training with rich labels from $\mathcal{D}_{\mathcal{S}}$ and a few labeled samples from $\mathcal{D}_{\mathcal{T}}$. For the fact that we can easily collect such additional labels on the target data in real-world applications, SSDA has the potential to render the adaptation problem more practical and promising in comparison to UDA.

Broadly, most contemporary approaches Ganin et al. (2016); Jiang et al. (2020); Kim & Kim (2020); Yoon et al. (2022) handle the SSDA task based on two domain shift assumptions, where $\mathcal{X}$ and $\mathcal{Y}$ respectively denote the samples and their corresponding labels: (i) *Covariate Shift*, $P(\mathcal{X}|\mathcal{D} = \mathcal{D}_{\mathcal{S}}) \neq P(\mathcal{X}|\mathcal{D} = \mathcal{D}_{\mathcal{T}})$; (ii) *Conditional Shift*, $P(\mathcal{Y}|\mathcal{X}, \mathcal{D} = \mathcal{D}_{\mathcal{S}}) \neq P(\mathcal{Y}|\mathcal{X}, \mathcal{D} = \mathcal{D}_{\mathcal{T}})$, refers to the difference of conditional label distributions of cross-domain data. Intuitively, one straightforward solution for SSDA is to learn the common features to mitigate the domain shift issues. Further quantitative analyses, however, indicate that the model trained with supervision on a few labeled target samples and labeled source data can just ensure partial cross-domain feature alignment Kim & Kim (2020). That is, it only aligns the features of labeled target samples and their correlated nearby samples with the corresponding feature clusters in the source domain.

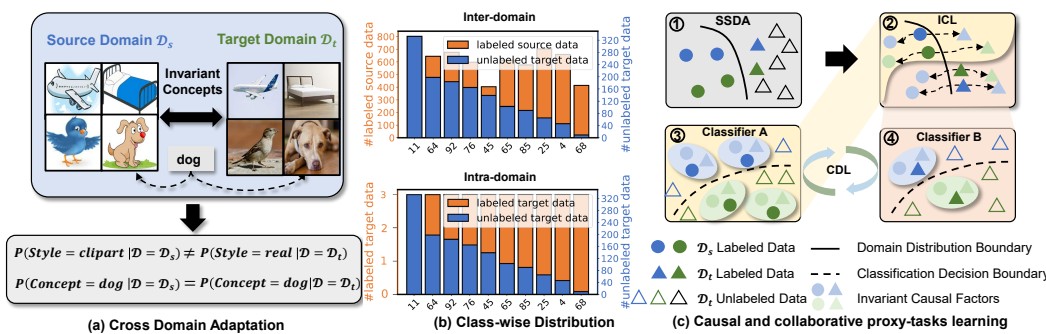

Figure 1: (a) Four DA cases ("Clipart" → "Real"). (b) Class-wise distribution of source domain and target domain. (c) A simplified version that indicates how our proposed model facilitates the SSDA.

To systematically study the SSDA problem, we begin by asking two fundamental questions, **Q1**:*What properties should a robust DA model have?* To answer this question, we first present a DA example in Figure 1(a), which suggests that the image "style" in $\mathcal{D} = \mathcal{D}_{\mathcal{T}}$ is drastically different from the $\mathcal{D} = \mathcal{D}_{\mathcal{S}}$. A classifier trained on the source domain may fail to predict correct labels even though the "concept" (*e.g.*, plane) is invariant with a similar outline. The truth is that the minimalist style features being invariant in "clipart" domain plays a critical factor in the trained classifier, which may consequently downplay the concept features simply because they are not as invariant as style features. Importantly, such an observation reveals the fundamental reason of the two domain shift assumptions, *i.e.*, $P(Style = clipart|\mathcal{D} = \mathcal{D}_{\mathcal{S}}) \neq P(Style = real|\mathcal{D} = \mathcal{D}_{\mathcal{T}})$. Therefore, a robust DA model needs to distinguish the invariant concept features in $\mathcal{X}$ across domains from the changing style. **Q2**: *How to maximally exploit the target domain supervision for robust SSDA?* As discussed, supervised learning on the few target labels cannot guarantee the global cross-domain feature alignment, which hurts the model generalization for invariant learning. A commonly known approach in this few labeled setting, semi-supervised learning (SSL), uses a trained model on labeled data to predict convincing pseudo-labels for the unlabeled data. This approach relies on the ideal assumption that the labeled and unlabeled data have the same marginal distribution of label over classes to generate pseudo-labels. However, Figure 1(b) indicates these distributions are different in both inter-domain and intra-domain. This may result in the imperfect label prediction that causes the well-known *confirmation bias* Arazo et al. (2020), affecting the model feature alignment capability. Further, in the SSDA setting, we have three sets of data, *i.e.*, source domain data, labeled and unlabeled target domain data. One single model for SSDA may be hard to generalize to the three sets with different label distributions. Thus, the premise of better utilization of labeled target samples is to mitigate undesirable bias and reasonably utilize the multiple sets. Summing up, these limitations call for reexamination of SSDA and its solutions.

To alleviate the aforementioned limitations, we propose a framework called **CA**usal collaborative proxy-tas**K**s l**E**arning (**CAKE**) which is illustrated in Figure 1(c). In the first step, we formalize the DA task using a causal graph. Then leveraging causal tools, we identify the "style" as the *confounder* and derive the invariant concepts across domains. In the subsequent steps, we build two classifiers based on the invariant concept to utilize rich information from cross-domain data for better SSDA. In this way, CAKE explicitly decomposes the SSDA into two proxy subroutines, namely *Invariant Concept Learning Proxy* (ICL) and *Collaboratively Debiasing Learning Proxy* (CDL). In ICL, we identify the key to robust DA is that the underlying concepts are consistent across domains, and the *confounder* is the style that prevents the model from learning the invariant concept ($C$) for accurate DA. Therefore, a robust DA model should be an invariant predictor $P(\mathcal{Y}|\hat{\mathcal{X}}, \mathcal{D} = \mathcal{D}_{\mathcal{T}}) = P(\mathcal{Y}|\hat{\mathcal{X}}, \mathcal{D} = \mathcal{D}_{\mathcal{S}}))$ under causal interventions. To address the problem, we devise a causal factor generator (CFG) that can produce concept-invariant samples $\hat{\mathcal{X}}$ with different style to facilitate the DA model to effectively learn the invariant concept. As such, our ICL may be regarded as an improved version of Invariant Risk Minimization (IRM) Arjovsky et al. (2019) for SSDA, which equips the model with the ability to learn the concept features that are invariant to styles. In CDL, with the invariant concept learning as the foundation, we aim to unleash the potential of three sets of cross-domain data for better SSDA. Specifically, we build two correlating and complementary pseudo-labeling based semi-supervised learning (SSL) classifiers for $\mathcal{D}_{\mathcal{S}}$ and $\mathcal{D}_{\mathcal{T}}$ with self-penalization. These two classifiers ensure that the mutual knowledge is exchanged to expand the number of "labeled" samples

in the target domain, thereby bridging the feature distribution gap. Further, to reduce the *confirmation bias* learned from respective labeled data, we adopt Inverse Propensity Weighting (IPW) Glynn & Quinn (2010) theory which aims to force the model to pay same attention to popular ones and tail ones in SSL models. Specifically, we use the prior knowledge of marginal distribution to adjust the optimization objective from P($\mathcal{Y}|\mathcal{X}$) to P($\mathcal{X}|\mathcal{Y}$) (Maximizing the probability of each $x \in \mathcal{X}$ with different $y \in \mathcal{Y}$ ) for unbiased learning. Thus, the negative impact caused by label distribution shift can be mitigated. Consequently, the two subroutines mutually boost each other with respect to their common goal for better SSDA.

Our contributions are three-fold: (1) We formalize the DA problem using causality and propose the explicitly invariant concept learning paradigm for robust DA. (2) To unleash the power of cross-domain data, we develop a collaboratively debiasing learning framework that effectively reduces the domain gap to enforce invariant prediction. (3) We extensively evaluate the proposed CAKE. The empirical results show that it outperforms SOTA approaches on the commonly used benchmarks.

## 2 DOMAIN ADAPTATION THROUGH CAUSAL LENSES: FINDING THE DEVIL

We shall start by grounding the domain adaptation (DA) in a causal framework to illustrate the key challenges of cross-domain generalization. As discussed in introduction, given data $\mathcal{X}$ and their labels $\mathcal{Y}$, the main difficulty of DA is that the extracted representation from $\mathcal{X}$ is no longer a strong visual cue for sample label in another domain. To study this issue in-depth, we first make the following assumption:

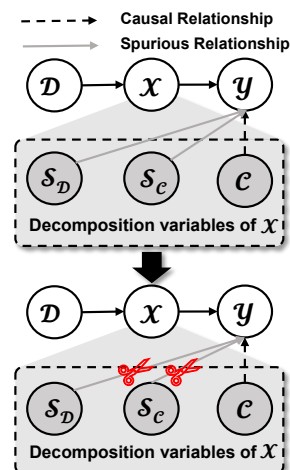

Figure 2: Causal graph of DA.

**Assumption 1 (Disentangled Variables).** *Data $\mathcal{X}$ can be disentangled into concept $\mathcal{C}$, cross-domain style $\mathcal{S}_\mathcal{C}$ and intra-domain style $S_I$ variables which are mutually independent, i.e., $\mathcal{X} = (\mathcal{C}, \mathcal{S}_\mathcal{C}, \mathcal{S}_\mathcal{I})$, where $\mathcal{C} \perp\!\!\!\perp \mathcal{S}_\mathcal{C} \perp\!\!\!\perp \mathcal{S}_\mathcal{I}$. Only concept $\mathcal{C}$ is relevant for the true label $\mathcal{Y}$ of $\mathcal{X}$, i.e., style changing is concept-preserving.*

Under this assumption, we abstract the DA problem into a causal graph (Figure 2 ). In this figure, $\mathcal{D}$ represents the Domain (*e.g.*, $\mathcal{D}_\mathcal{S}$ or $\mathcal{D}_\mathcal{T}$), while $\mathcal{S}_\mathcal{I}$ (*e.g.*, different appearance of concept in same domain) and $\mathcal{S}_\mathcal{C}$ (*e.g.*, different background of concept cross-domain) are the nuisance variables that confound $\mathcal{Y}$. The absence of any style changing is irrelevant for true label $\mathcal{Y}$. $\mathcal{C}$ is the invariant concept which contains directly causal relationships with $Y$. Therefore, the causal graph reveals the fundamental reasons for distinguishing issues across domains, *i.e.*, the cross/intra-domain style serves as the confounding variables that influence the $\mathcal{X} \to \mathcal{Y}$.

$$P(\mathcal{Y}|\mathcal{C}, \mathcal{D} = \mathcal{D}_\mathcal{S}) = P(\mathcal{Y}|\mathcal{C}, \mathcal{D} = \mathcal{D}_\mathcal{T}) \quad and \quad P(\mathcal{Y}|\mathcal{S}, \mathcal{D} = \mathcal{D}_\mathcal{S}) \neq P(\mathcal{Y}|\mathcal{S}, \mathcal{D} = \mathcal{D}_\mathcal{T})$$
$$\implies P(\mathcal{Y}|\mathcal{X}, \mathcal{D} = \mathcal{D}_\mathcal{S}) \neq P(\mathcal{Y}|\mathcal{X}, \mathcal{D} = \mathcal{D}_\mathcal{T}), \quad \forall \mathcal{S} \in \{\mathcal{S}_\mathcal{C}, \mathcal{S}_\mathcal{I}\}, \tag{1}$$

*The "devil" for DA problem could be style confounders $\mathcal{S}_\mathcal{C}$ and $\mathcal{S}_\mathcal{I}$ in that they prevent the model from learning the concept-invariant causality $\mathcal{X} \to \mathcal{Y}$[1]. From the causal theoretical view, such confounding effect can be eliminated by statistical learning with causal intervention Pearl et al. (2000). Putting all these observations together, we now state the main theorem of the paper.*

**Theorem 1 (Causal Intervention)**. *Under the causal graph in Figure 2 and Assumption 1, we can conclude that under this causal model, performing interventions on $\mathcal{S}_\mathcal{C}$ and $\mathcal{S}_\mathcal{I}$ does not change the $P(\mathcal{Y}|\mathcal{X})$. Thus, in DA problem, the causal effect $P(\mathcal{Y}|do(\mathcal{X})$[2]$, \mathcal{D} = \mathcal{D}_\mathcal{T})$ can be computed as:*

$$P(\mathcal{Y}|do(\mathcal{X}), \mathcal{D} = \mathcal{D}_\mathcal{T}) = P(\mathcal{Y}|do \underbrace{(\mathcal{C}, \mathcal{S}_\mathcal{C}, \mathcal{S}_\mathcal{I})}_{\text{Disentangled Variables}}, \mathcal{D} = \mathcal{D}_\mathcal{T}) = \sum_{\mathcal{D} \in \{\mathcal{D}_\mathcal{S}, \mathcal{D}_\mathcal{T}\}} \sum_{\hat{s}_\mathcal{C} \sim \mathcal{S}_\mathcal{C}} \sum_{\hat{s}_\mathcal{I} \sim \mathcal{S}_\mathcal{I}}$$

$$P(\mathcal{Y}|\mathcal{C}, \hat{s}_\mathcal{C}, \hat{s}_\mathcal{I}, \mathcal{D}) P(\mathcal{C}, \hat{s}_\mathcal{C}, \hat{s}_\mathcal{I}, \mathcal{D}) \approx \sum_{\hat{x} \sim \hat{\mathcal{X}}} P(\mathcal{Y}|\mathcal{X}, \hat{\mathcal{X}} = \hat{x}) P(\mathcal{X}, \hat{\mathcal{X}} = \hat{x}), \tag{2}$$

---

[1]While this assumption may not be true in all settings, we believe that the single image classification can be approximated by this assumption. More discussion about this assumption is in the appendix.

[2]$P(\mathcal{Y}|do(\mathcal{X}), \mathcal{D} = \mathcal{D}_\mathcal{T})$ uses the *do*-operator Glymour et al. (2016). Given random variables $\mathcal{X}, \mathcal{Y}$, we write $P(\mathcal{Y} = y|do(\mathcal{X} = x))$ to indicate the probability of $Y = y$ when we intervene and set $\mathcal{X}$ to be $x$.

where $\hat{\mathcal{X}}$ are the invariant causal factors with the same concepts of $\mathcal{X}$ but contain different cross/intra-domain styles, *i.e.*, invariant concept-aware samples. Realistically, $\hat{\mathcal{X}}$ is often a large set due to the multiple style combinations. This may block the model's computational efficiency according to Eq. 2 and hard to obtain such numerous causal factors. However, it is non-trivial to personally determine the $\hat{\mathcal{X}}$ size to study the deconfounded effect. We employ a compromise solution that significantly reduces the $\hat{\mathcal{X}}$ size to a small number for causal intervention.

## 3 CAKE: CAUSAL AND COLLABORATIVE PROXY-TASKS LEARNING

This section describes the CAKE for Semi-Supervised Domain Adaptation (SSDA) based on the studied causal and collaborative learning. We shall present each module and its training strategy.

### 3.1 PROBLEM FORMULATION

In the problem of SSDA, we have access to a set of labeled samples $\mathcal{S}_l = \{(x_{sl}^{(i)}, y_{sl}^{(i)})\}_{i=1}^{\mathcal{N}_s}$ i.i.d from source domain $\mathcal{D}_\mathcal{S}$. And the goal of SSDA is to adapt a learner to a target domain $\mathcal{D}_\mathcal{T}$, of which the training set consists of two sets of data: a set of unlabeled data $\mathcal{T}_u = \{(x_{tu}^{(i)})\}_{i=1}^{\mathcal{N}_u}$ and a small labeled set $\mathcal{T}_l = \{(x_{tl}^{(i)}, y_{tl}^{(i)})\}_{i=1}^{\mathcal{N}_l}$. Typically, we have $\mathcal{N}_l \leq \mathcal{N}_u$ and $\mathcal{N}_l \ll \mathcal{N}_s$. We solve the problem by decomposing the SSDA task into two proxy subroutines: Invariant Concept Learning (ICL) and Collaboratively Debiasing Learning (CDL). Such subroutines are designed to seek a robust learner $\mathcal{M}(\cdot; \Theta)$ which performs well on test data from the target domain:

$$\underbrace{\mathcal{M}(\cdot; (\Theta_\mathcal{I}, \Theta_\mathcal{C}))}_{\text{Learner CAKE}} : \underbrace{\mathcal{M}_\mathcal{I}((\hat{\mathcal{S}}_l, \hat{\mathcal{T}}_l, \hat{\mathcal{T}}_u) | (\mathcal{S}_l, \mathcal{T}_u, \mathcal{T}_l)); \Theta_\mathcal{I})}_{\text{ICL Proxy Subroutine}} \leftrightarrow \underbrace{\mathcal{M}_\mathcal{C}((\mathcal{T}_p | (\mathcal{S}_l, \hat{\mathcal{S}}_l, \mathcal{T}_l, \hat{\mathcal{T}}_l, \mathcal{T}_u, \hat{\mathcal{T}}_u)); \Theta_\mathcal{C})}_{\text{CDL Proxy Subroutine}} \quad (3)$$

where $\mathcal{M}_I$ and $\mathcal{M}_C$ indicate the ICL model parameterized by $\Theta_\mathcal{I}$ and the CDL model parameterized by $\Theta_\mathcal{C}$ respectively. In ICL proxy, $\mathcal{M}_I(\cdot; \Theta_I)$ learns the causal factors $(\hat{\mathcal{S}}_l, \hat{\mathcal{T}}_l, \hat{\mathcal{T}}_u)$ for $\mathcal{D}_\mathcal{S}$ and $\mathcal{D}_\mathcal{T}$ in unsupervised learning paradigm, aiming to generate the invariant causal factors and use Eq. 2 to remove the confounding effect. In CDL aspect, we construct two pseudo labeling-based SSL techniques: $(\mathcal{S}_l, \hat{\mathcal{S}}_l) \rightarrow \mathcal{T}_u$ and $(\mathcal{T}_l, \hat{\mathcal{T}}_l) \rightarrow \mathcal{T}_u$, aiming at utilizing all the training data possible to bridge the feature discrepancy under the premise of invariant concept learning.

### 3.2 INVARIANT CONCEPT LEARNING PROXY

As we discussed in Section 2, the key to robust DA is to eliminate the spurious correlations between styles ($\mathcal{S}_\mathcal{C}$ and $\mathcal{S}_\mathcal{I}$) and label $\mathcal{Y}$. To tackle this problem, we propose an *approximate* solution to kindly remove the confounding effect induced by $\mathcal{S}_\mathcal{C}$ and $\mathcal{S}_\mathcal{I}$. In detail, we develop the two invariant causal factor generators that can produce the causal factors $\hat{X}$ with $\mathcal{C}$. Next, we propose the Invariant Concept Learning (ICL) loss function, which forces the backbone (*e.g.*, ResNet-34  He et al. (2016) ) to focus on learning concepts that are invariant across a set of domains.

#### 3.2.1 INVARIANT CAUSAL FACTOR GENERATOR

Achieving the invariant concept-aware $\hat{X}$ is challenging due to the fact that supervised signals are missing or expensive to obtain. Thus, we resort to the unsupervised learning paradigm, designing two causal factor generators $C^{fg}(\cdot)=C_\mathcal{C}^{fg}(\cdot)$ (cross-domain) and $C_\mathcal{I}^{fg}(\cdot)$ (intra-domain) to achieve $\hat{\mathcal{X}}$ for $\mathcal{D}_\mathcal{S}$ and $\mathcal{D}_\mathcal{T}$ without the reliance on the supervised signals. Take $\mathcal{D} = \mathcal{D}_\mathcal{S}$ as an example, the invariant causal factors of $\mathcal{S}_l$ is given by $\hat{\mathcal{S}}_l = \{\hat{\mathcal{S}}_l^t, \hat{\mathcal{S}}_l^s\} = \{C_\mathcal{C}^{fg}(\mathcal{S}_l), C_\mathcal{I}^{fg}(\mathcal{S}_l)\}$ w.r.t $\mathcal{S}_\mathcal{C}$ and $\mathcal{S}_\mathcal{I}$:

**Cross-domain Causal Factor.** $\hat{\mathcal{S}}_l^t$ are generated by $\mathcal{N}_g$ GAN-based techniques Creswell et al. (2018), enabling the source concept to be preserved during the cross-domain conversion process. By considering the huge domain discrepancy, we optimize the style transfer loss as follows:

$$\min_{G_{st}^k} \max_{D_t^k} \mathcal{L}_{st}^k(\cdot; \Theta_F) = \mathbb{E}_{x_{sl} \sim \mathcal{S}_l, x_t \sim [\mathcal{T}_u; \mathcal{T}_l]}[\log D_t^k(x_t) + \log(1 - D_t^k(G_{st}^k(x_{sl})))$$
$$+ \mathcal{L}_{cyc}^k(x_{sl}, x_t; \Theta_F) + \mathcal{L}_{idt}^k(x_{sl}, x_t; \Theta_F)], k = \underset{\{i \in 1, \cdots, N_g\}}{\arg\min} \mathcal{L}_{st}^i(\cdot; \Theta_F), \quad (4)$$

where $[\cdot;\cdot]$ represents the union of two inputs, $D_t$ is the discriminator to distinguish the original source of the latent vector if from $\mathcal{D}_\mathcal{T}$. $\mathcal{L}_{cyc}^k$ and $\mathcal{L}_{idt}^k$ are the cycle and identity loss Zhu et al. (2017). $G_{st}^k$ is $k^{th}$ $C_\mathcal{C}^{fg}$. Through min-max adversarial training, the domain style-changing samples are obtained.

**Intra-domain Causal Factor.** We utilize the image augmentations as intra-domain style interventions, *e.g.*, modifying color temperature, brightness, and sharpness. We randomly adjust these image properties as our mapping function to change the intra-domain style for $\mathcal{D}_\mathcal{S}$ with invariant concept.

Thus, the invariant causal factors $\hat{\mathcal{S}}_l = \{\hat{\mathcal{S}}_l^t, \hat{\mathcal{S}}_l^s\}$ are produced. Correspondingly, for the target domain, $\hat{\mathcal{T}}_l$ and $\hat{\mathcal{T}}_u$ are also obtained in the generating learning strategy.

### 3.2.2 ICL Optimization Objective

After obtaining a set of invariant concept-aware samples $\mathcal{S}_l$ for source domain $\mathcal{D}_\mathcal{S}$, the goal of the proposed ICL can thus be formulated as the following optimization problem:

$$\min_{\Theta_\mathcal{I}^b, \Theta_\mathcal{I}^c} \mathcal{L}_{icl}(\cdot; (\Theta_\mathcal{I}^b, \Theta_\mathcal{I}^c)) = \mathbb{E}_{(\tilde{x}_{sl}, y_{sl}) \sim [\mathcal{S}_l, \hat{\mathcal{S}}_l^t, \hat{\mathcal{S}}_l^s]}[\mathcal{L}_{cls}(\Phi(\tilde{x}_{sl}; \Theta_\mathcal{I}^b), y_{sl}; \Theta_\mathcal{I}^c) + \lambda_{ir} \cdot \mathcal{L}_{ir}(\cdot; \Theta_\mathcal{I}^b)]$$

$$s.t. \quad \Theta_\mathcal{I}^b = \arg\min_{\hat{\Theta}_\mathcal{I}^b} \sum_{x_{sl} \sim \mathcal{S}_l} (\sum_{\mathcal{G} \in \{\mathcal{C}, \mathcal{I}\}} d(\Phi(x_{sl}), f(C_\mathcal{G}^{fg}(x_{sl}))) + d(\Phi(C_\mathcal{C}^{fg}(x_{sl})), \Phi(C_\mathcal{I}^{fg}(x_{sl}))) \tag{5}$$

where $\Theta_\mathcal{I}^b$ and $\Theta_\mathcal{I}^c$ are learnable parameters for the backbone and classifier, respectively. $\Phi(\tilde{x}_{sl}; \Theta_\mathcal{I}^b)$ is the backbone extracting feature from $\tilde{x}_{sl}$. $\lambda_{ir}$ is the trade-off parameter and $d(\cdot)$ is the euclidean distance between two inputs. $\mathcal{L}_{cls}(\Phi(\tilde{x}_{sl}; \Theta_\mathcal{I}^b), y_{sl}; \Theta_\mathcal{I}^c)$ is the cross-entropy loss for classification. To further access the concept-invariant learning effect, we develop the invariant regularization loss $\mathcal{L}_{ir}(\cdot; \Theta_\mathcal{I}^b)$ through a regularizer. We feed the $\mathcal{S}_l, \hat{\mathcal{S}}_l^s, \hat{\mathcal{S}}_l^t$ into the backbone network and explicitly enforcing them have invariant prediction, *i.e.*, $\text{KL}(P(\mathcal{Y}|\mathcal{S}_l), P(\mathcal{Y}|\hat{\mathcal{S}}_l^s), P(\mathcal{Y}|\hat{\mathcal{S}}_l^t)) \le \epsilon^3$. Such regularization is converted to an entropy minimization process McLachlan (1975), which encourages the classifier to focus on the *domain-invariant concept* and downplay the *domain-variant style*. The key idea of ICL similarly corresponds to the principle of **invariant risk minimization** (IRM) which aims to model the data representation for invariant predictor learning. More discussion about IRM and ICL is in the appendix.

### 3.3 Collaboratively Debiasing Learning Proxy

After invariant concept-aware samples generation, we obtain the $\hat{\mathcal{S}}_l$, $\hat{\mathcal{T}}_l$ and $\hat{\mathcal{T}}_u$. Next, we will elaborate on how to utilize the advantages of the extra supervised signals of target domain data $\mathcal{T}_l$ over the UDA setting. We introduce the Collaboratively Debiasing Learning framework (CDL) based on the robust DA setting with causal intervention. Specifically, we construct two SSL models: $\mathcal{M}_\mathcal{C}^s(\cdot; \Theta_\mathcal{C}^s)$ w.r.t $\{\mathcal{S}_l, \hat{\mathcal{S}}_l \text{ and } \mathcal{T}_u\}$ and $\mathcal{M}_\mathcal{C}^t(\cdot; \Theta_\mathcal{C}^t)$ w.r.t $\{\mathcal{T}_l, \hat{\mathcal{T}}_l \text{ and } \mathcal{T}_u\}$ as two complementary models with the same network architecture, which can cooperatively and mutually produce the pseudo-labels for each other to optimize the parameters Chen et al. (2011); Qiao et al. (2018). For instance, pseudo-label $\bar{y}$ of $x_{tu} \sim \mathcal{T}_u$ from $\mathcal{M}_\mathcal{C}^s(\cdot; \Theta_\mathcal{C}^s)$ is given by:

$$\bar{y} = \arg\max_{\tilde{y}}(P(\tilde{y}|x_{tu}; \Theta_\mathcal{C}^s) > \tau_s), \ where \ \tilde{y} = \mathcal{M}_\mathcal{C}^t(x_{tu}; \Theta_\mathcal{C}^t) \ if \ P(\tilde{y}|x_{tu}; \Theta_\mathcal{C}^t) > \tau_t \tag{6}$$

where $\tau_s$ and $\tau_t$ are the predefined threshold for pseudo-label selection. We will further elaborate on the two components in CDL, namely, a debiasing mechanism and a self-penalization technique. Without loss of generality, we describe the components using one of the SSL models $\mathcal{M}_\mathcal{C}^s(\cdot; \Theta_\mathcal{C}^s)$.

### 3.3.1 Confirmation Bias Eliminating Mechanism

The ultimate objective of most SSL frameworks is to minimize a risk, defined as the expectation of a particular loss function over a labeled data distribution $(\mathcal{X}, \mathcal{Y}) \sim \mathcal{S}_l$ Van Engelen & Hoos (2020).

---

[3]Note that any distance measure on distributions can be used in place of the Kullback-Leibler (KL) divergence Van Erven & Harremos (2014)

Therefore, the optimization problem generally becomes finding $\Theta_{\mathcal{S}}^s$ that minimizes the SSL risk.

$$\min_{\Theta_{\mathcal{C}}^s} \mathcal{R}(\cdot; \Theta_{\mathcal{C}}^s) = \mathbb{E}_{(x_{sl}, y_{sl}) \sim \mathcal{S}_l}[Ent_s((x_{sl}, y_{sl}); \Theta_{\mathcal{C}}^s)] + \mathbb{E}_{x_{tu} \sim \mathcal{T}_u}[\lambda_u \cdot Ent_u(x_{tu}; \Theta_{\mathcal{C}}^s)],$$

$$s.t. \quad \Theta_{\mathcal{C}}^s = \arg\max_{\hat{\Theta}_{\mathcal{C}}^s} \sum_{(x_{sl}, y_{sl}) \sim \mathcal{S}_l} \log P_s(y_{sl} | x_{sl}; \hat{\Theta}_{\mathcal{C}}^s)] \tag{7}$$

where $\lambda_u$ is the fixed scalar hyperparameter denoting the relative weight of the unlabeled loss. $Ent_s(\cdot)$ and $Ent_t(\cdot)$ are the cross-entropy loss function for labeled data $\mathcal{S}_l$ and unlabeled data $\mathcal{T}_u$.

**Proposition 1 (Origin of Confirmation Bias).** *SSL methods estimate the model parameters $\Theta_{\mathcal{C}}^s$ via maximum likelihood estimation according to labeled data $(\mathcal{X}, \mathcal{Y}) \sim \mathcal{S}_l$. Thus, the confirmation bias $\mathcal{B}_c$ in SSL methods is generated from the fully observed instances, namely labeled data.*

Under this proposition, the unbiased SSL learner should be impartial for less popular data (*e.g.*, tail samples $\mathcal{X}_t$) and popular ones (*e.g.*, head samples $\mathcal{X}_h$), *i.e.*, $P(\mathcal{X}_h | \mathcal{Y}) = P(\mathcal{X}_t | \mathcal{Y})$. Inspired by the inverse propensity weighting Glynn & Quinn (2010) theory, we get the unbiased theorem for SSL.

**Theorem 2 (Unbiased SSL Label Propagator).** *The optimization parameter $\Theta_{\mathcal{C}}^s$ for SSL model should be taken same attention for all the labeled data,* i.e.*, turn maximizing $\sum_{x_{sl} \in \mathcal{S}_l} \log P(x_{sl} | y_{sl}); \Theta_{\mathcal{C}}^s)$ (Complete proof in Appendix.).*

$$\Theta_{\mathcal{C}}^s = \arg\max_{\hat{\Theta}_{\mathcal{C}}^s} \sum_{(x_{sl}, y_{sl}) \sim \mathcal{S}_l} \log P(y_{sl} | x_{sl}; \hat{\Theta}_{\mathcal{C}}^s) = \arg\max_{\hat{\Theta}_{\mathcal{C}}^s} \sum_{x_{sl} \sim \mathcal{S}_l} \log P(x_{sl} | y_{sl}; \hat{\Theta}_{\mathcal{C}}^s) \cdot S_{IPW}(x_{sl}, y_{sl})$$

$$S_{IPW}(x_{sl}, y_{sl}) = \sum_{(x_{sl}, y_{sl}) \sim \mathcal{S}_l} P(y_{sl} | x_{sl}; \hat{\Theta}_{\mathcal{C}}^s) / (\log P(y_{sl} | x_{sl}; \Theta_C^s) - \log P(y_{sl}; \hat{\Theta}_{\mathcal{C}}^s)) \tag{8}$$

where $S_{IPW}(\cdot)$ is the Inverse Probability Weighting score. This formula can be understood as using the prior knowledge of marginal distribution $P(\mathcal{Y}; \Theta_{\mathcal{C}}^s)$ to adjust the optimization objectives for unbiased learning. To make practical use of this Eq. 20, we estimate $P(\mathcal{Y}; \Theta_{\mathcal{C}}^s, B_s, t)$ in each mini-batch training for error backpropagation at iteration $t$ with batch size $B_s$. It is noteworthy that we use a distribution moving strategy over all the iterations to reduce the high-variance estimation between time adjacent epochs. With the gradual removal of bias from the training process, the performance gap between classes also shrinks, and both popular and rare classes can be fairly treated.

### 3.3.2 SELF-PENALIZATION OF INDIVIDUAL CLASSIFIER

We also design a self-penalization that encourages the SSL model to produce more convincing pseudo-labels for exchanging peer classifier knowledge. Here, the negative pseudo-label indicates the most confident label (top-1 label) predicted by the network with a confidence lower than the threshold $\tau_s$. Since the negative pseudo-label is unlikely to be a correct label, we need to increase the probability values of all other classes except for this negative pseudo-label. Therefore, we optimize the output probability corresponding to the negative pseudo-label to be close to zero. The objective of self-penalization is defined as follows:

$$\min_{\Theta_{\mathcal{C}}^s} \mathcal{L}_{sp}(\cdot; \Theta_{\mathcal{C}}^s) = \mathbb{E}_{(x_{tu}, y_{tu}) \sim \hat{\mathcal{T}}_u}[\mathbb{1}(\max(P(y_{tu} | x_{tu}; \Theta_{\mathcal{C}}^s) < \tau_s)) \cdot y_{tu} \log(1 - P(y_{tu} | x_{tu}; \Theta_{\mathcal{C}}^s))] \tag{9}$$

Such self-penalization is able to encourage the model to generate more faithful pseudo-labels with a high-confidence score, and hence improve the data utilization for better invariant learning.

## 4 EXPERIMENTS

### 4.1 DATASET AND SETTING

**Benchmark Datasets.** `DomainNet` is originally a multi-source domain adaptation benchmark. Following Saito et al. (2019) in its use for SSDA evaluation, we only select 4 domains, which are Real, Clipart, Painting, and Sketch (abbr. **R**, **C**, **P** and **S**), each of which contains images of 126 categories. `Office-Home` Venkateswara et al. (2017) benchmark contains 65 classes, with 12 adaptation scenarios constructed from 4 domains (*i.e.*, **R**: Real world, **C**: Clipart, **A**: Art, **P**: Product). `Office` Saenko et al. (2010) is a relatively small dataset contains three domains including DSLR, Webcam and Amazon (abbr. D, W and A) with 31 classes.

Table 1: Accuracy(%) comparison on `DomainNet` under the settings of 3-shot using Resnet34 as backbone networks. A larger score indicates better performance. Acronym of each model can be found in Section 4.1. We color each row as the best, second best, and third best.

| Method | R to C | R to P | P to C | C to S | S to P | R to S | P to R | Mean Accuracy |
|---|---|---|---|---|---|---|---|---|
| S+T | 60.8 | 63.6 | 60.8 | 55.6 | 59.5 | 53.3 | 74.5 | 61.2 |
| DANN Ganin et al. (2016) | 59.8 | 62.8 | 59.6 | 55.4 | 59.9 | 54.9 | 72.2 | 60.7 |
| MME Saito et al. (2019) | 72.2 | 69.7 | 71.7 | 61.8 | 66.8 | 61.9 | 78.5 | 68.9 |
| APE Kim & Kim (2020) | 76.6 | 72.1 | 76.7 | 63.1 | 66.1 | 67.8 | 79.4 | 71.7 |
| SSSD Yoon et al. (2022) | 75.9 | 72.1 | 75.1 | 64.4 | 70.0 | 66.7 | 80.3 | 72.1 |
| DECOTA Yang et al. (2021) | 80.4 | 75.2 | 78.7 | 68.6 | 72.7 | 71.9 | 81.5 | 75.6 |
| CDAC Li et al. (2021b) | 79.6 | 75.1 | 79.3 | 69.9 | 73.4 | 72.5 | 81.9 | 76.0 |
| PACL Li et al. (2020) | 79.0 | 77.3 | 79.4 | 70.6 | 74.6 | 71.6 | 82.4 | 76.4 |
| Baseline | 75.4 | 71.8 | 74.2 | 65.9 | 70.3 | 70.2 | 78.8 | 72.4 |
| **CAKE (Ours)** | 83.3 | 77.6 | 79.1 | 72.2 | 73.0 | 74.5 | 83.2 | 77.6 |

**Implementation Details.** We employ the ResNet-34 He et al. (2016) and VGG-16 Simonyan & Zisserman (2014) as the backbone model on `DomainNet` and `Office-Home`, respectively. We train CAKE with a SGD Bottou (2010) optimizer in all experiments. Besides, we use an identical set of hyperparameters ($B$=24, $M_o$=0.9, $L_r$, $\tau$=0.5, $T_{max}$=20,000, $\lambda_s$=1, $\lambda_u$=1, $\lambda_{ir}$=0.1, $\lambda_{sp}$=0.1). The causal factor generator $C_{\mathcal{C}}^{fg}(\cdot)$=CycleGan Zhu et al. (2017) and $C_{\mathcal{I}}^{fg}(\cdot)$=Image Augmentation, $\mathcal{M}_{\mathcal{C}}(\cdot; \Theta_{\mathcal{C}})$=Mixmatch Berthelot et al. (2019)) [4] across all datasets.

**Comparison of Methods.** For quantifying the efficacy of the proposed framework, we compare CAKE with previous SOTA SSDA approaches, including **MME** Saito et al. (2019), **DANN** Ganin et al. (2016), **BiAT** Jiang et al. (2020), **APE** Kim & Kim (2020), **DECOTA** Yang et al. (2021), **CDAC** Li et al. (2021b) and **SSSD** Yoon et al. (2022). More details of baselines are in the appendix.

## 4.2 EXPERIMENTAL RESULTS AND ANALYSES

**Comparison with SOTA Methods.** Table 1, and 7 (in appendix) summarize the quantitative three-shot results of our framework and baselines on `DomainNet` and `Office-Home`. The one-shot results and analysis are in the supplementary material. In general, irrespective of the adaptation scenario, CAKE achieves the best performance on almost all the metrics to SOTA on the two datasets. In particular, CAKE outperforms other baselines in terms of Mean Accuracy by a large margin (`DomainNet`: **1.2% ∼ 16.4%**, `Office-Home`: **3.3% ∼ 9.7%** and `Office`: **3.8% ∼ 12.0%**) for SSDA task. Notably, our baseline, a simplified variant of CAKE without causal intervention and debiasing operation also obtained comparable results compared with SOTA (-3.6%). These results both benefit from the carefully designed ICL and CDL proxy subroutines that demonstrate the superiority and generalizability of our proposed model.

**Individual Effectiveness of Each Component.** We conduct an ablation study to illustrate the effect of each component in Table 2, which indicates the following: Causal Inference is critical to boost SSDA (Row 5 *vs.* Row 6), which significantly contributes 2.4% and 1.9% improvement on `DomainNet` and `Office-Home`, respectively. Meanwhile, Row 1 indicates that it suffers from noticeable performance degradation without the bias-removed mechanism (Row 1) (-1.3% and -1.8%). Furthermore, the results of Row 3 and Row 4 severally show the performance improvement of the Invariant Regularization ($\mathcal{L}_{ir}$) and Self-penalization ($\mathcal{L}_{sp}$). Summing up, We can observe that the improvement of using either module alone is distinguishable. Combining all the superior components, our CAKE exhibits steady improvement over the baselines.

**Maximally Cross-domain Data Utilization.** Here, we evaluate the effectiveness of data utilization of the proposed method. Figure 3 (a) and (b) show the comparison between CAKE and baseline with respect to the top-1-accuracy, accuracy and number pseudo-labels on `DomainNet` (Real → Clipart).

Subscript $\mathcal{S}$ and $\mathcal{T}$ represent the learned trained on source domain $\mathcal{D}_{\mathcal{S}}$ or target domain $\mathcal{D}_{\mathcal{T}}$. During the learning iteration, we observe that the accuracy of CAKE increases much faster and smoother

---

[4]$B$, $M_o$, $L_r$ and $T_{max}$ refer to batch size, momentum, learning rate and max iteration in SGD optimizer. The $\mathcal{M}_{\mathcal{I}}$ and $\mathcal{M}_{\mathcal{C}}$ are orthogonal to other advanced style changing and SSL methods to boost SSDA further.

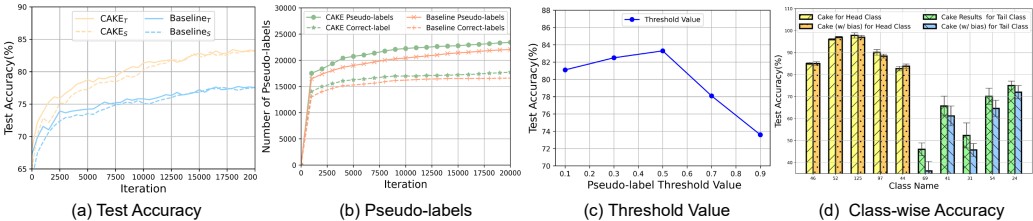

| (a) Test Accuracy | (b) Pseudo-labels | (c) Threshold Value | (d) Class-wise Accuracy |

Figure 3: **Analysis of cross-domain data utilization and debiasing mechanism of CAKE.** (a) and (b) depict the top-1-accuracy and correct pseudo-labels of CAKE and baseline within the first 200K iterations. (c) Cake's sensitivity to pseudo-label threshold $\tau$. (d) demonstrates the class-wise accuracy for head and tail classes in dataset produced by CAKE (w/o)/(w/) *confirmation bias*.

than baseline, and outperforms baseline by a large margin of accuracy. CAKE also produces more convincing pseudo-labels than baseline. These pseudo-labels can assist the SSDA in performing global domain alignment to decrease the intra-domain discrepancy for robust invariant concept learning. Apart from learning visualization, we also investigate the CAKE's sensitivity to the confidence threshold $\tau$ for assigning pseudo-labels. Figure 3 (c) empirically provides an appropriate threshold, *i.e.*, $\tau$=0.5, either increasing or decreasing this value results in a performance decay. What's more, we conducted the *cooperation vs. solo* ablation that verifies the power of collaborative learning in Table 3. The detached SSL model performs worse, demonstrating that training two correlated models achieves better adaptation as opposed to only aligning one of them, because collaborative learning allows both models to learn common knowledge from different domains that in turn facilitates invariant learning. The aforementioned observation and analysis verify the effectiveness of CAKE in being able to deeply mine the potential of cross-domain data, thereby achieving the SSDA improvement.

**Table 2:** Ablation study that showcases the impact of individual module.

| Method | DomainNet | Office-Home |
|---|---|---|
| -IWP Score | 75.3 | 73.6 |
| -Invariant Regularization | 76.7 | 74.9 |
| -Self-penalization | 77.0 | 74.8 |
| -Causal Intervention | 75.2 | 73.5 |
| **CAKE (Ours)** | **77.6** | **75.4** |

**Effect of Confirmation Bias Eliminating.** To build insights on the unbiased SSL in CAKE, we perform an in-depth analysis of the bias-eliminating mechanism in Figure 3(d). In this experiment, we randomly select 10 classes (5 head and 5 tail). The results suggest that CAKE and its variant CAKE (w/ bias) obtain a comparable performance on the head class. However, CAKE (w/ bias) fails to maintain the consistent superiority on the tail class while our approach does. (*e.g.*, tail class 69, CAKE: 46.0% , CAKE (w/ bias) : 36.2%). This phenomenon is reasonable since CAKE maintains unbiasedness to each class-wise sample by maximizing $P(\mathcal{X}|\mathcal{Y})$. As the labeled/unlabeled data share the same class distribution, the accuracy of the tail class can be improved. In contrast, CAKE (w/ bias) focuses more on the head class, which results in an unbalanced performance for all categories. These results empirically verified our theoretical analysis and the robustness of the debiasing mechanism, which provides a reliable solution that guarantees the mutual data knowledge to be exchanged from source and target aspects.

**Number of Invariant Causal Factors.** Figure 4(a) reports the SSDA results of different numbers of Invariant Causal Factors (ICFs) $\hat{\mathcal{X}}$ ($2 \times \mathcal{N}_g$) on DomainNet. Across all scenes, the best performance is usually achieved with $\mathcal{N}_g$ = 2, except for P $\rightarrow$ C. This ablation proves the ICFs of $\hat{\mathcal{X}}$ can be learned from a set of limited style-changing samples. Appropriately using these ICFs to conduct the deconfounded operation can effectively improve the SSDA performance.

**Table 3: Results of cooperation *vs. solo*.**

| Method | Domain | Office-Home |
|---|---|---|
| $\mathcal{M}_c^s$ | 70.6 | 67.4 |
| $\mathcal{M}_c^t$ | 68.3 | 65.8 |
| **CAKE (Ours)** | **77.6** | **75.4** |

**Grad-CAM Results of Causal Intervention.** We systematically present the explicit benefits of the invariant concept learning (ICL). Figure 4(b) visualizes the most influential part in prediction generated from Grad-CAM Selvaraju et al. (2017). It's rather clear to see that CAKE appropriately captures the invariant part of the concept while CAKE(w/o CI) failed. We also analyze the reason why CAKE performs better in these cases. For instance, the concept $C$="celling fan" has a complicated background, *i.e.*, style $\mathcal{S}$="cluttered". Without causal intervention, CAKE (w/o ICL) tends to focus

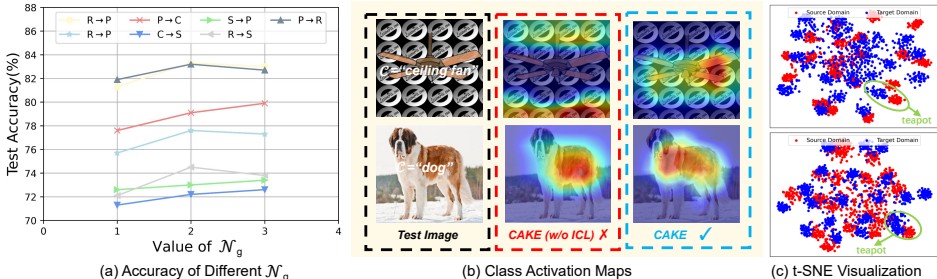

Figure 4: **In-depth analysis of CAKE.** (a) is the plot of invariant causal factor number $\mathcal{N}_g$ against accuracy(%). (b) Grad-CAM results of CAKE and CAKE(w/o ICL). (c) t-SNE plot of features.

on the irrelevant information of style $\mathcal{S}$= "cluttered", therefore predicting the wrong class. On the contrary, CAKE can attend to the vital image regions by learning the invariant concept $\mathcal{C}$= "celling fan" through the deconfounded mechanism.

**Cross-domain Feature Alignment.** We employ the t-SNE Van der Maaten & Hinton (2008) to visualize the feature alignment before/after training of adaptation scenarios $\mathcal{D}_\mathcal{S}$="Real" and $\mathcal{D}_\mathcal{T}$="Clipart" on `DomainNet`. We randomly select 1000 samples (50 samples per class). Our invariant concept learning focuses on making $\mathcal{X}_\mathcal{S}$ and $\mathcal{X}_\mathcal{T}$ alike. It can be observed that as the model optimization progresses, *e.g.*, $\mathcal{C}$="teapot", the target features gradually converge toward target cluster cores. Each cluster in the target domain also gradually moves closer to its corresponding source cluster cores, showing a cluster-wise feature alignment effect. This provides an intuitive explanation of how our CAKE alleviates the domain shift issue.

## 5 RELATED WORK

**Semi-supervised Domain Adaptation.** Semi-supervised domain adaptation (SSDA) Saito et al. (2019); Qin et al. (2020); Jiang et al. (2020); Li & Hospedales (2020); Kim & Kim (2020); Li et al. (2021b); Yoon et al. (2022) address the domain adaptation problem where some target labels are available. However, these techniques mainly rely on the two domain shift assumptions of *Covariate Shift* and *Conditional Shift* to conduct SSDA. Such assumptions present intuitive solutions but lack a solid theoretical explanation for the effectiveness of SSDA, which hinders their further development. Thus we develop the CAKE, which decomposes the SSDA as two proxy subroutines with causal theoretical support and reveals the fundamental reason of the two domain shift assumptions.

**Invariant Risk Minimization.** Recently, the notion of invariant prediction has emerged as an important operational concept in the machine learning field, called IRM Rosenfeld et al. (2020); Arjovsky et al. (2019). IRM proposes to use group structure to delineate between different environments where the aim is to minimize the classification loss while also ensuring that the conditional variance of the prediction function within each group remains small. In DA, this idea can be studied by learning classifiers that are robust against domain shifts Li et al. (2021a) but still has the *Covariate Shift* issue. Therefore, we propose the CAKE that enforces the model to learn the local disentangled invariant-concepts rather than the global invariant-features across domains, thus facilitating the SSDA.

**Causality in DA.** There are some causality study in DA community. Glynn & Quinn (2010) considered domain adaptation where both the distribution of the covariate and the conditional distribution of the target given the covariate change across domains. Gong et al. (2016) consider the target data causes the covariate, and an appropriate solution is to find conditional transferable components whose conditional distribution given the target is invariant after proper location-scale transformations, and estimate the target distribution of the target domain. Different from the two causal DA handle the DA task that only deals with the *Conditional Shift* issue, we also consider the *Covariate Shift*, which presents a improved IRM view for SSDA.

## 6 CONCLUSION

We first propose a causal framework to pinpoint the causal effect of disentangled style variables, and theoretically explain what characteristics should a robust domain adaptation model have. We next discuss the maximal training data utilization and present a collaboratively debiasing learning framework to make use of the training data to boost SSDA effectively. We believe that CAKE serves as a complement to existing literature and provides new insights to the domain adaptation community.

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

This is the Appendix for "CAKE: CAusal and collaborative proxy-tasKs lEarning for Semi-Supervised Domain Adaptation". Table 4 summarizes the abbreviations and the symbols used in the main paper.

**Table 4:** Abbreviations and symbols used in the main paper.

| Abbreviation/Symbol | Meaning |
|---|---|
| | *Abbreviation* |
| DA | Domain Adaptation |
| UDA | Unsupervised Domain Adaptation |
| SSDA | Semi-supervised Domain Adaptation |
| SSL | Semi-supervised Learning |
| CAKE | Causal and collaborative proxy-tasks learning |
| ICL | Invariant Concept Learning |
| CDL | Collaboratively Debiasing Learning |
| ICF | Invariant Causal Factor |
| ICFG | Invariant Causal Factor Generator |
| IRM | Invariant Risk Minimization |
| | *Symbol in Theory* |
| $\mathcal{D}_{\mathcal{S}}$ | Source Domain |
| $\mathcal{D}_{\mathcal{T}}$ | Target Domain |
| $\mathcal{X}$ | Image Data |
| $\mathcal{Y}$ | True Label |
| $\mathcal{C}$ | Concept |
| $\mathcal{S}_{\mathcal{C}}$ | Cross-domain Style |
| $\mathcal{S}_{\mathcal{I}}$ | Intra-domain Style |
| $\hat{\mathcal{X}}$ | Invariant Causal Factors |
| $2 \times \mathcal{N}_g$ | The Number of Causal Factors |
| | *Symbol in Algorithm* |
| $\mathcal{S}_l$ | Source Domain Data |
| $\mathcal{T}_l$ | Labeled Target Domain Data |
| $\mathcal{T}_u$ | Unlabeled Target Domain Data |
| $\mathcal{M}_{\mathcal{I}}$ | ICL Proxy Subroutine |
| $\mathcal{M}_{\mathcal{C}}$ | CDL Proxy Subroutine |
| $\Theta_{\mathcal{I}}$ | Parameters of $\mathcal{M}_{\mathcal{I}}$ |
| $\Theta_{\mathcal{C}}$ | Parameters of $\mathcal{M}_{\mathcal{C}}$ |
| $\hat{\mathcal{S}}_l$ | Causal factors for $\mathcal{D}_{\mathcal{S}}$ |
| $\hat{\mathcal{T}}_l, \hat{\mathcal{T}}_u$ | Causal Factors for $\mathcal{D}_{\mathcal{T}}$ |
| $\mathcal{L}_{icl}$ | Loss of Invariant Concept Learning |
| $\mathcal{L}_{cdl}$ | Loss of Collaboratively Debiasing Learning |
| $\mathcal{L}_{ir}$ | Invariant Regularization Loss |
| $\mathcal{L}_{sp}$ | Self-penalization Loss |

This appendix is organized as follows:

- Section 7 provides the proof about the disentangled $X$ causal intervention, invariant risk minimization, unbiased eliminating mechanism and further discussion of Assumption 1.

- Section 8 provides the method details of proposed CAKE.

- Section 9 reports more experimental settings of datasets, baselines, implementation details and training process of CAKE.

- Section 10 shows the additional experiments on `DomainNet` and `Office-Home` Venkateswara et al. (2017) to verify the effectiveness of CAKE.
- Section 11 lists the limitations of this paper.

# 7 PROOF AND DERIVATION

This section derives the disentangled causal intervention and the inverse probability weighting theory for the confirmation of unbiased eliminating mechanism.

## 7.1 DISENTANGLED $\mathcal{X}$ CAUSAL INTERVENTION

We will first provide a brief introduction to the preliminaries of disentangled $\mathcal{X}$ causal intervention. Real-world observations, according to physicists, are the result of a mix of independent physical rules. This also applies to causal inference Higgins et al. (2018), *i.e.*, the laws are denoted as disentangled generative factors, such as shape, color, and position. Let the $\mathcal{X}$ represent the image data, each $\mathcal{X}$ can be disentangled into concept $\mathcal{C}$, cross-domain style $\mathcal{S}_{\mathcal{C}}$ and intra-domain style $\mathcal{S}_{\mathcal{I}}$ variables which are mutually independent, *i.e.*, a triplet $\mathcal{X} = (\mathcal{C}, \mathcal{S}_{\mathcal{C}}, \mathcal{S}_{\mathcal{I}})$, where $\mathcal{C} \perp\!\!\!\perp \mathcal{S}_{\mathcal{C}} \perp\!\!\!\perp \mathcal{S}_{\mathcal{I}}$. Correspondingly, the invariant causal factors of $X$ are given by $\hat{\mathcal{X}}$ ( style mapping results of $\mathcal{X}$), where $\mathcal{S} \in \{\mathcal{S}_{\mathcal{C}}, \mathcal{S}_{\mathcal{I}}\}$ are different from $\mathcal{X}$. Only concept $\mathcal{C}$ is relevant for the true label $\mathcal{Y}$ of $\mathcal{X}$, *i.e.*, style changing is concept-preserving. In other words, there is a set of independent causal mechanisms $\phi \colon \mathcal{S} \to \mathcal{X}, \hat{\mathcal{X}}$, generating images from $\mathcal{S}$. To study how can we get the $\hat{\mathcal{X}}$, we leverage the assumption of disentangled variables based on Higgins' definition of **disentangled representation** Higgins et al. (2018). We state the definition as follows:

**Definition 1 (Group Action on Disentangled Variables).** *Let $\mathcal{G}$ be the group acting on $\mathcal{S}$, i.e, $g \cdot s \in \mathcal{S} \times \mathcal{S}$ transforms $s \in \mathcal{S}$, e.g., group element of "turn domain style, real to clipart" changing the semantic from "real" to "clipart". Suppose there is a direct product decomposition $\mathcal{G} = g_1 \times g_2 \times \cdots \times g_q$ and $\mathcal{S} = \mathcal{S}_1 \times \mathcal{S}_2 \times \cdots \times \mathcal{S}_q$, where $g_i$ acts on $S_i$ respectively. A feature representation is disentangled if there exists a group $\mathcal{G}$ acting on $\mathcal{X}$ such that:*

- **Theorem 3 ( Decomposable $\mathcal{X}$.)** *There is a decomposition $\mathcal{X} = \mathcal{X}_1 \times \mathcal{X}_2 \times \cdots \times X_q$, such that each $X_i$ is fixed by the action of all $g_j$, where $j \neq i$ and affected only by $g_i$, e.g., changing the "domain style" semantic in $\mathcal{S}$ does not affect the "concept" vector in $\mathcal{X}$.*

$$P(\mathcal{X}_j = concept | g_i \cdot \mathcal{X}_i) = P(\mathcal{X}_j = concept | \mathcal{X}_i) \tag{10}$$

- **Theorem 4 (Equivariant Semantic Changing.)** $\forall g \in \mathcal{G}, \forall s \in \mathcal{S}, f(g \cdot s) = g \cdot f(s)$, *e.g., the feature of the changed cross-domain style semantic: "real" to "clipart" in $\mathcal{S}$, is equivalent to directly change the style vector in $\mathcal{X}$ from "real" to "clipar".*

$$P(g \cdot s | \mathcal{S}) = P(g \cdot s | \mathcal{X}) \tag{11}$$

Under this **Theorem 3**, the disentangled representations are obtained by our **Assumption 1**, *i.e.*, $\mathcal{X} = (\mathcal{C}, \mathcal{S}_{\mathcal{C}}, \mathcal{S}_{\mathcal{I}})$. Compared to the previous definition of feature representation which is a static mapping, the disentangled representation in **Definition 1** is dynamic as it explicitly incorporates group representation Williams (2002), which is a homomorphism from group to group actions on a space, *e.g.*, $\mathcal{G} \to \mathcal{X} \times \mathcal{X}$, and it is common to use the feature space $\mathcal{X}$. What's more, **Theorem 4** indicates that performing group action of semantic changing (*e.g.*, style changing) $g$ on $S$ is equivariant for $\mathcal{S}$ and $\mathcal{X}$. Thus, $\hat{\mathcal{X}}$ can be obtained by performing different $g \in \mathcal{G}$ on $\mathcal{X}$.

Next, we can introduce the causality that allows computing how an outcome would have changed, had some variables taken different values, referred to as a causal intervention. As a prerequisite, $\hat{\mathcal{X}}$ should be calculated following the three steps of computing principles Pearl & Mackenzie (2018):

- *In abduction, all the invariant causal factors, i.e., $(\hat{\mathcal{X}} = \hat{x}_1, \hat{X} = \hat{x}_2, \cdots, \hat{\mathcal{X}} = \hat{x}_k)$ are inferred from $\mathcal{X}$ through $P(\hat{\mathcal{X}} | \mathcal{X})$ .*

- *In action, $\hat{\mathcal{X}} = \hat{x}_i$ is drawn from $P(\hat{\mathcal{X}} = \hat{x}_i | \mathcal{D} = \mathcal{D}_{\mathcal{S}})$ or $P(\hat{\mathcal{X}} = \hat{x}_i | \mathcal{D} = \mathcal{D}_{\mathcal{T}})$, while the values of other $\hat{\mathcal{X}}$ are fixed.*

- *In prediction, the modified $(\hat{\mathcal{X}} = \hat{x}_1, \hat{X} = \hat{x}_2, \cdots, \hat{X} = \hat{x}_k)$ is fed to the generative process $P(\mathcal{X}|\hat{\mathcal{X}})$ to obtain the output.*

More details can be found in Glymour et al. (2016). Based on the computing principles, we consider $\mathcal{G}$ as an embedded function Besserve et al. (2018), *i.e.*, a continuous injective function with continuous inversion, which generally holds for convolution-based networks as shown in Puthawala et al. (2020). $\hat{\mathcal{X}}$ are obtained through the generative process $\mathcal{G} : \mathcal{X} \rightarrow \hat{\mathcal{X}}$. $\hat{\mathcal{X}}$ with the invariant concept with $\mathcal{X}$ can be regarded as the causal factors to take a causal theoretical view for the domain adaptation problem.

**Proof of the Sufficient Condition.** Suppose that the representation is fully disentangled *w.r.t.* $\mathcal{G}$. By Definition 1, there exists subspace $\mathcal{X}_i \in \mathcal{X}$ affected only by the action of $g_i \in \mathcal{G}$. This part aims to prove the following sufficient condition: if $g_i$ intervenes $S_i$, the invariant causal factors(ICFs) are faithful when the $\mathcal{L}_{st}^i$ is smallest or the $i^{th}$ image augmentation have not change the concept. For a sample $x_{\mathcal{S}}$ from $\mathcal{D} = \mathcal{D}_{\mathcal{S}}$, let $g^{-1}(x_{\mathcal{S}}) = \mathcal{S} = (\mathcal{S}_1, \cdots \mathcal{S}_2, \cdots \mathcal{S}_k)$. We modify style by changing $g_i \in \mathcal{G}$ drawn from $P(g_i|\mathcal{D} = \mathcal{D}_{\mathcal{T}})$ (cross-domain style) or $P(g_i|\mathcal{D} = \mathcal{D}_{\mathcal{S}})$ (intra-domain style). Denote the modified style as $\hat{\mathcal{S}} = (\hat{\mathcal{S}}_1 \times \hat{\mathcal{S}}_2 \times \cdots \times \hat{\mathcal{S}}_k)$. Denote the sample with style $\hat{\mathcal{S}}$ as $\hat{x}_{\mathcal{S}}$. Given $g_i$ intervenes $\mathcal{S}$ corresponds to a counterfactual outcome when $\mathcal{S}_i$ is set to $\hat{\mathcal{S}}_i$ through intervention. Now as $g^{-1}(x_{\mathcal{S}}) = \mathcal{S}$, using the counterfactual consistency rule, we have $g_i(x_{\mathcal{S}}) = \hat{x}_{\mathcal{S}}$. As $\hat{x}_{\mathcal{S}}$ is faithful with the Counterfactual Faithfulness theorem Pearl et al. (2000), we prove that $g_i(x_{\mathcal{S}})$ is also faithful, *i.e.*, the smallest $\mathcal{L}_{st}^i$ or the $i^{th}$ image augmentation have not change the concept.

## 7.2 INVARIANT RISK MINIMIZATION

In a seminal work, Arjovsky et al. (2019) consider the question that data are collected from multiple envrionments with different distributions where spurious correlations are due to dataset biases. This part of spurious correlation will confuse model to build predictions on unrelated correlations rather than true causal relations. IRM estimates invariant and causal variables from multiple environments by regularizing on predictors to find data represenation matching for all environments.

Let $\mathcal{X}$ be the image space, $\mathcal{Z}$ and $\mathcal{Y}$ represent the be feature space and classification output space (*e.g.*, the set of all probabilities of belonging to each class), the feature extractor backbone $\Phi : \mathcal{X} \rightarrow \mathcal{Z}$ and the classifier $w : \mathcal{Z} \rightarrow \mathcal{Y}$. Let $\mathcal{E}_{tr}$ be a set of training environments, where each $e \in \mathcal{E}_{tr}$ is a set of images. Mathematically, IRM phrase these goals as the constrained optimization problem:

$$\min_{\Phi, w} \sum_{e \in \mathcal{E}_{tr}} \underbrace{\mathcal{R}^e(\cdot; \Phi)}_{\text{ERM Term}} + \lambda \cdot \underbrace{\left\| \nabla_{w|w=1.0} \mathcal{R}^e(w \cdot \Phi) \right\|^2}_{\text{Invariant Risk}} \tag{12}$$

where $\mathcal{R}^e(\cdot; \Phi)$ is the empirical classification risk (ERM) in the environment $e$, $w = 1.0$ is a scalar and fixed "dumm" classifier, the gradient norm penalty is used to measure the optimality of the dummy classifier at each environment $e$, and $\lambda \in [0, +\infty)$ is a regularizer balancing between predictive power, and the invariance of the predictor $1 \cdot \Phi(x)$.

In DA problem, IRM can be regarded as the classic ERM term (*e.g.*, classification loss) plus the invariant risk (*e.g.*, discrepancy between conditional distributions over the features) Li et al. (2021a). The invariant risk consider about the *Conditional Shift* assumption ($P(\mathcal{Y}|\mathcal{X}, \mathcal{D} = \mathcal{D}_{\mathcal{S}}) \neq P(\mathcal{Y}|\mathcal{X}, \mathcal{D} = \mathcal{D}_{\mathcal{S}})$) to learn the data representation, thereby learning the invariant predictor. However, IRM may not be the true savior for the DA task which still has two issues:

- **Covariate Shift.** As we discussed in section 1, the marginal feature distributions are different across domain, *i.e.*, $P(\mathcal{X}|\mathcal{D} = \mathcal{D}_{\mathcal{S}}) \neq P(\mathcal{X}|\mathcal{D} = \mathcal{D}_{\mathcal{T}})$. In SSDA task, there are only few labeled samples in $\mathcal{D}_{\mathcal{T}}$, it hard to measure the gap of marginal feature distribution between two domains. Thus, an alternative is consider the *Covariate Shift* that reduce the features discrepancy across domains. However, IRM is not sufficient to tackle this issue.

- **Spurious Correlation.** The learned global representation of an image still has noise style information rather than the fine-grained concept. Using such global representation may leave confused style information in feature space, resulting in inaccurate prediction.

Different from the IRM in DA, our ICL tackled the aforementioned issues at two points: 1) representing images from the source domain $\mathcal{D}_{\mathcal{S}}$ to target domain $\mathcal{D}_{\mathcal{T}}$ by invariant causal factor generator; 2) eliminating the spurious correlation of style and label by statistical learning with causal intervention.

In summary, our proposed CAKE not only addressed the two domain shift issue, but also enforced the model to learn the disentangled invariant-concepts across domains, which can boost the SSDA reasonably.

### 7.3 Inverse Probability Weighting Theory

The main content of this paper indicates the unbiased SSL learner should be impartial for less popular data (*e.g.*, tail samples $\mathcal{X}_t$) and popular ones (*e.g.*, head samples $\mathcal{X}_h$), *i.e.*, $P(\mathcal{X}_h|\mathcal{Y}) = P(\mathcal{X}_t|\mathcal{Y})$. Inspired by the inverse propensity weighting Glynn & Quinn (2010) theory, which introduces a weight for each training sample via its propensity score, which reflects how likely the label is observed (*e.g.*, its popularity). In this way, IPW makes up a pseudo-balanced dataset by duplicating each labeled data inversely proportional to its propensity—less popular samples should draw the same attention as the popular ones—a more balanced imputation. Thus, take $\mathcal{D} = \mathcal{D}_\mathcal{S}$ as an example, the optimization parameter $\Theta_\mathcal{C}^s$ for SSL model $\mathcal{M}_\mathcal{C}^s$ should be taken same attention for all the labeled data. In general SSL training, $\Theta_\mathcal{C}^s$ can be optimized as follows:

$$\Theta_\mathcal{C}^s = \arg\max_{\hat{\Theta}_\mathcal{C}^s} \sum_{(x_{sl},y_{sl})\sim\mathcal{S}_l} \log P(y_{sl}|x_{sl};\hat{\Theta}_\mathcal{C}^s) \tag{13}$$

$$= \arg\max_{\hat{\Theta}_\mathcal{C}^s} \sum_{(x_{sl},y_{sl})\sim\mathcal{S}_l} \log P(x_{sl}|y_{sl};\hat{\Theta}_\mathcal{C}^s) \cdot \frac{P(y_{sl};\hat{\Theta}_\mathcal{C}^s)}{P(x_{sl};\hat{\Theta}_\mathcal{C}^s)} \tag{14}$$

$$= \arg\max_{\hat{\Theta}_\mathcal{C}^s} \sum_{(x_{sl},y_{sl})\sim\mathcal{S}_l} \log P(x_{sl}|y_{sl};\hat{\Theta}_\mathcal{C}^s) \cdot P(y_{sl};\hat{\Theta}_\mathcal{C}^s) \tag{15}$$

$$= \arg\max_{\hat{\Theta}_\mathcal{C}^s} \sum_{(x_{sl},y_{sl})\sim\mathcal{S}_l} \log P(x_{sl}|y_{sl};\hat{\Theta}_\mathcal{C}^s) \cdot \frac{\log P(x_{sl}|y_{sl};\hat{\Theta}_\mathcal{C}^s) + \log P(y_{sl};\hat{\Theta}_\mathcal{C}^s)}{\log P(x_{sl}|y_{sl};\hat{\Theta}_\mathcal{C}^s)} \tag{16}$$

$$= \arg\max_{\hat{\Theta}_\mathcal{C}^s} \sum_{(x_{sl},y_{sl})\sim\mathcal{S}_l} \log P(x_{sl}|y_{sl};\hat{\Theta}_\mathcal{C}^s) \cdot \frac{\log \frac{P(y_{sl}|x_{sl};\hat{\Theta}_\mathcal{C}^s)P(x_{sl};\hat{\Theta}_\mathcal{C}^s)}{P(y_{sl};\hat{\Theta}_\mathcal{C}^s)} + \log P(y_{sl};\hat{\Theta}_\mathcal{C}^s)}{\log \frac{P(y_{sl}|x_{sl};\hat{\Theta}_\mathcal{C}^s)P(x_{sl};\hat{\Theta}_\mathcal{C}^s)}{P(y_{sl};\hat{\Theta}_\mathcal{C}^s)}} \tag{17}$$

$$= \arg\max_{\hat{\Theta}_\mathcal{C}^s} \sum_{(x_{sl},y_{sl})\sim\mathcal{S}_l} \log P(x_{sl}|y_{sl};\hat{\Theta}_\mathcal{C}^s) \cdot \frac{\log \frac{P(y_{sl}|x_{sl};\hat{\Theta}_\mathcal{C}^s)}{P(y_{sl};\hat{\Theta}_\mathcal{C}^s)} + \log P(y_{sl};\hat{\Theta}_\mathcal{C}^s)}{\log \frac{P(y_{sl}|x_{sl};\hat{\Theta}_\mathcal{C}^s)}{P(y_{sl};\hat{\Theta}_\mathcal{C}^s)}} \tag{18}$$

$$= \arg\max_{\hat{\Theta}_\mathcal{C}^s} \sum_{(x_{sl},y_{sl})\sim\mathcal{S}_l} \log P(x_{sl}|y_{sl};\hat{\Theta}_\mathcal{C}^s) \cdot \frac{\log P(y_{sl}|x_{sl};\hat{\Theta}_\mathcal{C}^s)}{\log P(y_{sl}|x_{sl};\hat{\Theta}_\mathcal{C}^s) - \log P(y_{sl};\hat{\Theta}_\mathcal{C}^s)} \tag{19}$$

$$\Rightarrow S_{IPW}(.) = \frac{\log P(y_{sl}|x_{sl};\hat{\Theta}_\mathcal{C}^s)}{\log P(y_{sl}|x_{sl};\hat{\Theta}_\mathcal{C}^s) - \log P(y_{sl};\hat{\Theta}_\mathcal{C}^s)} \tag{20}$$

where $S_{IPW}(\cdot)$ is the Inverse Probability Weighting score, $P(x_{xl};\hat{\Theta}_\mathcal{C}^s)$ is the sampling probability an empirical distribution which is thus constant. Thus, from (4) to (5) and (7) to (8) in Eq. 13, $P(x_{xl};\hat{\Theta}_\mathcal{C}^s)$ can be ignored. Therefore, we can turn maximizing $\arg\max_{\hat{\Theta}_\mathcal{C}^s} \sum_{(x_{sl},y_{sl})\sim\mathcal{S}_l} \log P(y_{sl}|x_{sl};\hat{\Theta}_C^s)$ to maximizing $\arg\max_{\Theta_\mathcal{C}^s} \sum_{(x_{sl},y_{sl})\sim\mathcal{S}_l} \log P(x_{sl}|y_{sl};\Theta_\mathcal{C}^s)$ according Eq. 13. This formula can be understood as using the prior knowledge of marginal distribution $P(\mathcal{Y};\Theta_\mathcal{C}^s)$ to adjust the optimization objectives for unbiased learning. Thus, the SSL model $\mathcal{M}_\mathcal{C}^s$ is unbiased to the class-wise sample by maximizing $\sum_{(x_{sl},y_{sl})\sim\mathcal{S}_l} \log P(x_{sl}|y_{sl};\Theta_\mathcal{C}^s)$, thereby eliminating the undesirable confirmation bias.

### 7.4 Further Discussion of Assumption 1

We would like to clarify the further explanation of our proposed assumption 1 (Disentangled Variables). Besides, we also give other intuition of this assumption for other tasks.

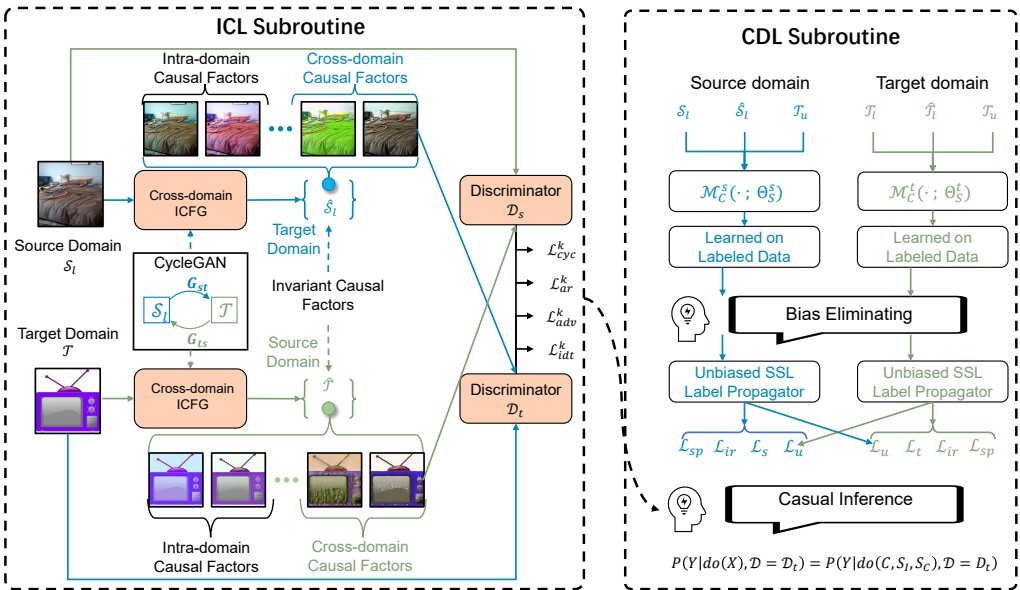

Figure 5: Overview of CAKE. In ICL proxy, $\mathcal{M}_{\mathcal{I}}(\cdot; \Theta_{\mathcal{I}})$ first learns the causal factors for $\mathcal{D}_{\mathcal{S}}$ and $\mathcal{D}_{\mathcal{T}}$ in unsupervised learning paradigm, aiming to generate the invariant causal factors and use causal intervention to remove the confounding effect. In CDL aspect, we construct two pseudo labeling-based SSL techniques: $(\mathcal{S}_l, \hat{\mathcal{S}}_l) \to \mathcal{T}_u$ and $(\mathcal{T}_l, \hat{\mathcal{T}}_l) \to \mathcal{T}_u$, aiming at utilizing all the training data possible to bridge the feature discrepancy under the premise of invariant "concept" learning.

- **Single classification task.** We have already noted that this assumption may not be true in all settings, but we believe that many image settings can be approximated. For example, in most of the single classification tasks, a given image (*e.g.*, a dog on the lawn) only has one label. The annotators tend to focus on the most important region (the dog) which can be regarded as the concept to give the label. The dog with other colors or appearance can be regarded as the intra-domain style . The dog in other domain (*e.g.*, clipart domain) with different backgrounds can refer to the cross-domain style . In other words, and are confounders that interfere the model to predict the true label given the image.

- **Other complex vision tasks.** For these tasks, *e.g.*, multi-label task Li et al. (2006), visual question answer Antol et al. (2015), visual captioning Vinyals et al. (2015), this assumption may not be applicable. For instance, an image describes "a dog and a cat on the lawn". When the image is classified as a dog, besides the style confounders , the cat is also an extraneous factor called object confounder. Nevertheless, we also investigated this scenario in and the results systematically show the robustness of the proposed CAKE.

# 8 METHOD DETAILS

This section presents the method details of the proposed CAKE. Figure. 5 illustrates the overview of our CAKE framework that contains two proxy subroutines: Invariant Concept Learning (ICL) and Collaboratively Debiasing Learning (CDL). Next, we will elaborate on the details of cross-domain style transfer and bias eliminating mechanism in practice.

## 8.1 CROSS-DOMAIN STYLE TRANSFER

The invariant concept samples with cross-domain style transferring is generated from CycleGAN Zhu et al. (2017) [5]. Here we provide the architectures of generator and discriminator in Table 5.

---

[5]https://github.com/junyanz/pytorch-CycleGAN-and-pix2pix

**Table 5:** Abbreviations and the symbols used in the main paper.

| Generator | Discriminator |
|---|---|
| ReflectionPad (3) | Conv2D (4,64) |
| Conv2D (7,64) | LeakyReLU (0.2) |
| InstanceNorm ReLU | Conv2D (4,128) |
| Conv2D (3,128) | InstanceNorm |
| InstanceNorm ReLU$\times$ 2 | LeakyReLU (0.2) |
| Conv2D (3,256) | Conv2D (4,256) |
| InstanceNorm ReLU | InstanceNorm |
| ResNetBlock $\times$ 2 | LeakyReLU (0.2) |
| InstanceNorm ReLU | Conv2D (4,512) |
| ConvTranspose2D (3,64) | InstanceNorm |
| InstanceNorm ReLU | LeakyReLU (0.2) |
| ReflectionPad (3) | Conv2D (4,1) |
| Conv2D (7,3) | |
| Tanh | |

**Loss of Cross-domain Invariant Causal Factor Generator.** The CycleGAN technique transforms the images to enable the preservation of source concepts during the cross-domain conversion process. Take $\mathcal{D}_\mathcal{S} \to \mathcal{D}_\mathcal{T}$ as an example, we develop $\mathcal{N}_g$ CycleGANs, each of them consisting of three-loss parts to conduct the cross-domain style transfer.

$$\arg\min_{\Theta_I} \mathcal{L}_{st}^k = \arg\min_{\Theta_I} (\lambda_{adv} \cdot \mathcal{L}_{adv}^k + \lambda_{cyc} \cdot \mathcal{L}_{cyc}^k + \lambda_{idt} \cdot \mathcal{L}_{idt}^k), \quad k = \underset{\{i \in 1, \cdots, \mathcal{N}_g\}}{\arg\min} \mathcal{L}_{st}^i \quad (21)$$

where $\mathcal{L}_{adv}^k$, $\mathcal{L}_{cyc}^k$ and $\mathcal{L}_{idt}^k$ are adversarial loss, cycle loss and identity loss, respectively. $\lambda_{adv}$, $\lambda_{cyc}$ and $\lambda_{idt}$ correspond to their trade-off parameters. Specifically, the $\mathcal{L}_{adv}^k$, $\mathcal{L}_{cyc}^k$ and $\mathcal{L}_{idt}^k$ can be calculated as follows:

$$\mathcal{L}_{adv}^k = \underset{x_t \sim (\mathcal{T}_u, \mathcal{T}_l)}{\mathbb{E}} [\log D_t^k(x_t)] + \underset{x_{sl} \sim \mathcal{S}_l}{\mathbb{E}} [\log(1 - D_t^k(G_{st}^k(x_{sl})))] \quad (22)$$

$$\mathcal{L}_{cyc}^k = \underset{x_{sl} \sim \mathcal{S}_l}{\mathbb{E}} [||G_{ts}^k(G_{st}^k(x_{sl})) - x_{sl}||] + \underset{x_t \sim (\mathcal{T}_u, \mathcal{T}_l)}{\mathbb{E}} [||G_{st}^k(G_{ts}^k(x_t)) - x_t||] \quad (23)$$

$$\mathcal{L}_{idt}^k = \underset{x_{sl} \sim \mathcal{S}_l}{\mathbb{E}} [||G_{ts}^k(x_{sl}) - x_{sl}||] + \underset{x_t \sim (\mathcal{T}_u, \mathcal{T}_l)}{\mathbb{E}} [||G_{st}^k(x_t) - x_t||] \quad (24)$$

where $||\cdot||$ denote the *L1* norm. $D_t$ is the discriminator to distinguish the origin source of the latent vector if from $\mathcal{D}_\mathcal{T}$. $G_{st}^k$ and $G_{ts}^k$ correspond to the $k^{th}$ $\mathcal{D}_\mathcal{S} \to \mathcal{D}_\mathcal{T}$ and $\mathcal{D}_\mathcal{T} \to \mathcal{D}_\mathcal{S}$ cross-domain invariant causal factor generator (ICFG), respectively.

## 8.2 INTRA-DOMAIN STYLE TRANSFER

For the intra-domain style changing factors, we utilize the data augmentations as intra-domain style interventions,*e.g.*, color temperature and sharpness according to the cross-domain style changing samples. The code of the intra-domain style transfer can be found in our online project. Thus, the invariant causal factors $\hat{\mathcal{S}}_l = \{\hat{\mathcal{S}}_l^t, \hat{\mathcal{S}}_l^s\}$ are produced. Correspondingly, for the target domain, $\hat{\mathcal{T}}_l$ and $\hat{\mathcal{T}}_u$ are obtained in the generating learning strategy.

## 8.3 INVARIANT CONCEPT LEARNING

After obtaining a set of invariant concept-aware samples $\mathcal{S}_l$ for source domain $\mathcal{D}_\mathcal{S}$, we design the ICL loss function which has two aspects as follows:

$$
\min_{\Theta_\mathcal{I}^b, \Theta_\mathcal{I}^c} \mathcal{L}_{icl}(\cdot; (\Theta_\mathcal{I}^b, \Theta_\mathcal{I}^c)) = \mathcal{L}_{cls} + \lambda_{ir} \cdot \mathcal{L}_{ir}(\cdot; \Theta_\mathcal{I}^b)
$$

$$
\min_{\Theta_\mathcal{I}^b} \mathcal{L}_{icl}(\cdot; \Theta_\mathcal{I}^b) = \sum_{x_{sl} \sim \mathcal{S}_l} \left( \sum_{\mathcal{G} \in \{\mathcal{C}, \mathcal{I}\}} d(\Phi(x_{sl}), f(C_\mathcal{G}^{fg}(x_{sl}))) + d(\Phi(C_\mathcal{C}^{fg}(x_{sl})), \Phi(C_\mathcal{I}^{fg}(x_{sl}))) \right) \quad (25)
$$

where $\Theta_\mathcal{I}^b$ and $\Theta_\mathcal{I}^c$ are learnable parameters for the backbone and classifier, respectively. $\Phi(\tilde{x}_{sl}; \Theta_\mathcal{I}^b)$ is the backbone extracting feature from $\tilde{x}_{sl}$. $\lambda_{ir}$ is the trade-off parameter and $d(\cdot)$ is the euclidean distance between two inputs. $\mathcal{L}_{cls}(\Phi(\tilde{x}_{sl}; \Theta_\mathcal{I}^b), y_{sl}; \Theta_\mathcal{I}^c)$ is the cross-entropy loss for classification. To further access the concept-invariant learning effect, we develop the invariant regularization loss $\mathcal{L}_{ir}(\cdot; \Theta_\mathcal{I}^b)$ through a regularizer. Such regularization is converted to an entropy minimization process McLachlan (1975), which encourages the classifier to focus on the *domain-invariant concept* and downplay the *domain-variant style*.

## 8.4 CONFIRMATION BIAS ELIMINATING MECHANISM

According to complete proof of IPW theory in section.7.3, we aim to find the optimal $\tilde{\Theta}_\mathcal{C}^s$ that maximizing $\sum_{(x_{sl}, y_{sl}) \sim \mathcal{S}_l} \log P(x_{sl}|y_{sl}; \Theta_\mathcal{C}^s)$ to implement the debiasing SSL model learning.

$$
\tilde{\Theta}_\mathcal{C}^s = \arg\max_{\Theta_\mathcal{C}^s} \sum_{(x_{sl}, y_{sl}) \sim \mathcal{S}_l} \log P(x_{sl}|y_{sl}; \Theta_\mathcal{C}^s) = \arg\max_{\Theta_\mathcal{C}^s} \sum_{(y_{sl}, x_{sl}) \sim \mathcal{S}_l} \frac{\log P(y_{sl}|x_{sl}; \Theta_\mathcal{C}^s)}{S_{IPW}(.)} \quad (26)
$$

To make practical use of this Eq. 20, we estimate $P(Y; \Theta_C^s, B_s, t)$ in each mini-batch training for error backpropagation at iteration $t$ with batch size $B_s$. It is noteworthy that we use a distribution moving strategy over all the iterations to reduce the high-variance estimation between time adjacent epochs. The details as below:

$$
\sum_{i=1}^{B_s} (M_o \cdot \hat{P}(y_{sl}^{(i)}; \Theta_\mathcal{C}^s, B_s, t - B_s) + (1 - M_o) \cdot P(y_{sl}^{(i)}; \Theta_\mathcal{C}^s, B_s, t)) \rightarrow \sum_{i=1}^{B_s} \hat{P}(y_{sl}^{(i)}; \Theta_\mathcal{C}^s, B_s, t)
$$
$$(27)$$

where $B_t$ is the batch size and $M_o$ is a momentum coefficient, $\hat{P}(\cdot)$ is the re-estimated prior. With the gradual removal of bias from the training process, the performance gap between classes also shrinks, and both popular and rare classes can be fairly treated.

## 8.5 OBJECTIVE FUNCTION OF CDL

The full objective function of CDL has three parts, supervised loss $\mathcal{L}_s(\cdot; \Theta_\mathcal{C})$, unsupervised loss $\mathcal{L}_u(\cdot; \Theta_\mathcal{C})$ and self-penalization loss $\mathcal{L}_{sp}(\cdot; \Theta_\mathcal{C})$:

$$
\min_{\Theta_\mathcal{C}} \mathcal{L}_{cdl}(\cdot; \Theta_\mathcal{C}) = \mathbb{E}[\lambda_s \cdot \mathcal{L}_s(\cdot; \Theta_\mathcal{C}) + \lambda_u \cdot \mathcal{L}_u(\cdot; \Theta_\mathcal{C}) + \lambda_{sp} \cdot \mathcal{L}_{sp}(\cdot; \Theta_\mathcal{C})], \quad \Theta_\mathcal{C} \in \{\Theta_\mathcal{C}^s, \Theta_\mathcal{C}^t\} \quad (28)
$$

where $\lambda_s$, $\lambda_u$ and $\lambda_{sp}$ denote the pre-defined hyper-parameters.

## 9 EXPERIMENTAL SETTINGS

**Implementation Details.** Our work can be checked at the **Anonymous Link** [6], . Algorithm 1 and 2 presents the pseudocode of training and inference process of CAKE. We use the PyTorch Paszke et al. (2019) deep learning framework to conduct all our experiments on $8\times$ V100 GPUs and $8\times$ 2080Ti GPUs. We employ the ResNet-34 He et al. (2016) and VGG-16 Simonyan & Zisserman (2014) (We also report the results of ResNet-34 on `Office-Home` ) as the backbone model on `DomainNet` [7]

---

[6]https://anonymous.4open.science/r/Cake-A1B0

[7]http://ai.bu.edu/M3SDA/

**Table 6:** Complete list of hyper-parameters.

| Hyperparameters | Notation | Value |
|---|---|---|
| Labeled Data Batch-size | $B_s$ | 24 |
| Confidence Threshold | $\tau$ | 0.5 |
| Momentum | $M_o$ | 0.9 |
| Learning Rate | $L_r$ | 0.001 |
| Max Iteration | $T_{max}$ | 20000 |
| HP of Supervised l Loss | $\lambda_s$ | 1 |
| HP of Unsupervised Loss | $\lambda_u$ | 1 |
| HP of Invariant Regularization Loss | $\lambda_{ir}$ | 0.1 |
| HP of Self-penalization Loss | $\lambda_{sp}$ | 0.1 |

and `Office-Home` [8], respectively. We train CAKE with a standard stochastic gradient descent (SGD) Bottou (2010) optimizer in all experiments. We follow Saito et al. (2019) to replace the last linear layer with a $K$-way cosine classifier (*e.g.*, K = 126 for DomainNet) and train it at a fixed temperature (0.05 in all our setting). Besides, we use an identical set of hyperparameters ($B$=24, $M_o$=0.9, $L_r$, $\tau$=0.5, $T_{max}$=20,000) [9] across all datasets. We utilize the Mixmatch Berthelot et al. (2019) [10] as the semi-supervised learning model, the basic loss function for CAKE consists of two cross-entropy loss terms: a supervised loss $\mathcal{L}_s$ applied to labeled data and an unsupervised loss $\mathcal{L}_u$.

**Comparison of Methods.** For quantifying the efficacy of the proposed framework, we compare CAKE with previous SOTA SSDA approaches, including **MME** Saito et al. (2019), **DANN** Ganin et al. (2016), **BiAT** Jiang et al. (2020), **APE** Kim & Kim (2020), **DECOTA** Yang et al. (2021), **ELP** Inoue et al. (2018), **CDAC** Li et al. (2021b) and **SSSD** Yoon et al. (2022). We also present a simplified version of CAKE as the baseline.

We compare the results of CAKE with a wide range of baselines, including early works and recent SOTA models on this task:

- **Baseline** is a simplified version of CAKE without causal intervention, bias eliminating mechanism, invariant regularization and self-penalization.
- **MME** Saito et al. (2019) first proposed to solve SSDA by aligning the features from both domains by means of adversarial learning.
- **DANN** Ganin et al. (2016) augmented the model with few standard layers and a new gradient reversal layer based on the features that cannot be discriminated between the source and target domains.
- **ELP** Inoue et al. (2018) designed a framework with domain transfer and pseudo labeling to generate instance-level annotations for the target domain.
- **BiAT** Jiang et al. (2020) devised a bidirectional strategy with an adaptive adversarial model and an entropy-penalized virtual adversarial model to guide the direction of generating adversarial examples.
- **APE** Kim & Kim (2020) addressed the intra-domain discrepancy issue via attraction, perturbation, and exploration schemas.
- **DECOTA** Yang et al. (2021) decomposed SSDA into an SSL and UDA problem as two models to bridge the gap and exchange expertise between the source and target domains.
- **CDAC** Li et al. (2021b) developed an adversarial adaptive clustering loss to guide the model training towards grouping the features of unlabeled target data into clusters and further performing cluster-wise feature alignment across domains.

---

[8]https://www.hemanthdv.org/officeHomeDataset.html

[9]$B$, $M_o$, $L_r$ and $T_{max}$ refer to batch size, momentum, learning rate and max iteration in SGD optimizer.

[10]https://github.com/YU1ut/MixMatch-pytorch

---

**Algorithm 1:** CAKE: Causal Multi-proxy Subroutine Learning Framework

---

1    **Input**: Training data $\mathcal{S}_l$ from source Domain $D_\mathcal{S}$, $\mathcal{T}_l$ and $\mathcal{T}_u$ from target domain $\mathcal{D}_\mathcal{T}$, pre-trained classifiers $\mathcal{M}_\mathcal{C}(\cdot; \Theta_\mathcal{C}^s)$ and $\mathcal{M}_\mathcal{C}(\cdot; \Theta_\mathcal{C}^t)$ with parameters $\Theta_C^s$ and $\Theta_\mathcal{C}^t$, respectively;

2    **Output**: Invariant causal factors of $\hat{\mathcal{S}}_l, \hat{\mathcal{T}}_l, \hat{\mathcal{T}}_u$, fine-tuned classifier $\mathcal{M}_\mathcal{C}(\cdot; \Theta_\mathcal{C}^t)$;

3    **Initialization**: Randomly initialize the parameters $\{\Theta_{st}^k\}_{k=1}^{N_g}$ of Cross-domain ICFGs $\{G_{st}^k\}_{k=1}^{N_g}$;

4    **repeat**

5      **if** *Cross-domain ICFG model in CAKE have not trained* **then**

6        **repeat**

7          Randomly sample a minibatch;

8          Train cross domain ICFGs $\{G_{st}\}_{i=1}^k$ with unpaired $\mathcal{S}_l, \mathcal{T}_l$ and $\mathcal{T}_u$ with cross-domain style transfer loss $\mathcal{L}_{st}$;

9          Update $\{\Theta_{st}^k\}_{k=1}^{N_g}$ using Eq. 4;

10      **until** *Convergence*;

11    **until** *Convergence*;

12    Image augmentations for $\mathcal{S}_l, \mathcal{T}_l$ and $\mathcal{T}_u$;

13    Generate $\hat{\mathcal{S}}_l = \{\hat{\mathcal{S}}_l^t, \hat{\mathcal{S}}_l^s\}, \hat{\mathcal{T}}_l = \{\hat{\mathcal{T}}_l^t, \hat{\mathcal{T}}_l^s\}, \hat{\mathcal{T}}_u = \{\hat{\mathcal{T}}_u^t, \hat{\mathcal{T}}_u^s\}$;

14    Initialize $\mathcal{M}_\mathcal{C}(\cdot; \Theta_\mathcal{C}^s)$ and $\mathcal{M}_\mathcal{C}(\cdot; \Theta_\mathcal{C}^t)$;

15    **repeat**

16      Randomly sample a minibatch from $\{\mathcal{S}_l, \hat{\mathcal{S}}_l\}$ and $\{\mathcal{T}_l, \hat{\mathcal{T}}_l\}$;

17      Eliminate confirmation bias $\mathcal{B}_c$ using Eq. 8;

18      Obtain pseudo-labels using Eq. 6;

19      Calculate CDL loss of $\mathcal{L}_s, \mathcal{L}_u$ and $\mathcal{L}_{sp}$;

20      Calculate $P(\mathcal{Y}|do(\mathcal{X}), \mathcal{D} = \mathcal{D}_\mathcal{S})$ and $P(\mathcal{Y}|do(\mathcal{X}), \mathcal{D} = \mathcal{D}_\mathcal{T})$ for $\mathcal{M}_\mathcal{C}(\cdot; \Theta_\mathcal{C}^s)$ and $\mathcal{M}_\mathcal{C}(\cdot; \Theta_\mathcal{C}^t)$ using Eq. 2;

21      Calculate ICL loss of $\mathcal{L}_{cls}$ and $\mathcal{L}_{ir}$ using Eq.25;

22      Update parameters $\Theta_\mathcal{C}^s$ and $\Theta_\mathcal{C}^t$;

23    **until** *Convergence*;

24    **return** *Classifiers* $\mathcal{M}_\mathcal{C}(\cdot; \Theta_\mathcal{C}^s)$ *and* $\mathcal{M}_\mathcal{C}(\cdot; \Theta_\mathcal{C}^t)$.

---

**Algorithm 2:** CAKE Inference

---

1    **Input**: Test data $\mathcal{T}_u$ from target domain $\mathcal{D}_\mathcal{T}$, fine-tuned classifier $\mathcal{M}_\mathcal{C}(\cdot; \Theta_\mathcal{C}^t)$ with parameters $\Theta_\mathcal{C}^t$;

2    **Output**: Predicted label $\hat{y}_t$;

3    Sample $x_{tu} \sim \mathcal{T}_u$;

4    Sample invariant causal factors $\{\hat{x}_{tu}^i\}_{i=1}^{2N_g}$ of $x_{tu}$;

5    Measure deconfounded effect using Eq. 2;

6    **return** $\hat{y}_{tu} \leftarrow \text{argmax } P(\mathcal{Y}|do(\mathcal{X} = x_{tu}), \mathcal{D} = \mathcal{D}_\mathcal{T})$.

---

- **SSSD** Yoon et al. (2022) exploited the rich sample-to-sample relations using self-distillation to fill the domain gap, thus facilitating the adaptation.

- **IRM** Arjovsky et al. (2019) estimates nonlinear, invariant, causal predictors from multiple training environments, to enable out-of-distribution generalization.

- **LIRR** Li et al. (2021a) simultaneously learning invariant representations and risks across domains, leading to a bound minimization algorithm for SSDA.

- **PACL** Li et al. (2020) uses a data augmentation-based technique to produce highly perturbed images to mitigate overfitting. The consistency alignment module is incorporated into the framework, which enforces consistency regularization on the classifier.

**Table 7:** Accuracy on `Office-Home` (%) for three-shot setting with 4 domains, using VGG-16. A larger score indicates better performance. We color each row as the best , second best , and third best .

| Method | R to C | R to P | R to A | P to R | P to C | P to A | A to P | A to C | A to R | C to R | C to A | C to P | MEAN |
|---|---|---|---|---|---|---|---|---|---|---|---|---|---|
| S+T | 49.6 | 78.6 | 63.6 | 72.7 | 47.2 | 55.9 | 69.4 | 47.5 | 73.4 | 69.7 | 56.2 | 70.4 | 62.9 |
| DANN Ganin et al. (2016) | 56.1 | 77.9 | 63.7 | 73.6 | 52.4 | 56.3 | 69.5 | 50.0 | 72.3 | 68.7 | 56.4 | 69.8 | 63.9 |
| MME Saito et al. (2019) | 56.9 | 82.9 | 65.7 | 76.7 | 53.6 | 59.2 | 75.7 | 54.9 | 75.3 | 72.9 | 61.1 | 76.3 | 67.6 |
| APE Kim & Kim (2020) | 56.0 | 81.0 | 65.2 | 73.7 | 51.4 | 59.3 | 75.0 | 54.4 | 73.7 | 71.4 | 61.7 | 75.1 | 66.5 |
| DECOTA Yang et al. (2021) | 59.9 | 83.9 | 67.7 | 77.3 | 57.7 | 60.7 | 78.0 | 54.9 | 76.0 | 74.3 | 63.2 | 78.4 | 69.3 |
| ELP Inoue et al. (2018) | 57.1 | 83.2 | 67.0 | 76.3 | 53.9 | 59.3 | 75.9 | 55.1 | 76.3 | 73.3 | 61.9 | 76.1 | 68.0 |
| Baseline | 60.4 | 82.7 | 66.6 | 76.7 | 58.2 | 59.8 | 76.3 | 55.9 | 76.2 | 73.7 | 63.0 | 76.0 | 68.8 |
| **CAKE (Ours)** | 62.4 | 87.0 | 70.8 | 80.6 | 61.5 | 64.3 | 81.4 | 58.6 | 79.5 | 77.3 | 65.9 | 81.6 | 72.6 |

**Table 8:** Accuracy(%) comparison on `Office` under the settings of 1-shot and 3-shot using Alexnet as backbone networks.

| Method | W to A | | D to A | | MEAN | |
|---|---|---|---|---|---|---|
| | one-shot | three-shot | one-shot | three-shot | one-shot | three-shot |
| S+T | 50.4 | 61.2 | 50.0 | 62.4 | 50.2 | 61.8 |
| DANN Ganin et al. (2016) | 57.0 | 64.4 | 54.5 | 65.2 | 55.8 | 64.8 |
| MME Saito et al. (2019) | 57.2 | 67.3 | 55.8 | 67.8 | 56.5 | 67.6 |
| BiAT Jiang et al. (2020) | 57.9 | 68.2 | 54.6 | 68.5 | 56.3 | 68.4 |
| APE Kim & Kim (2020) | - | 67.6 | - | 69.0 | - | 68.3 |
| CDAC Li et al. (2021b) | 63.4 | 70.1 | 62.8 | 70.0 | 63.1 | 70.0 |
| CAKE(Ours) | 66.9 | 74.5 | 66.0 | 73.1 | 66.5 | 73.8 |

## 10 ADDITIONAL EXPERIMENTAL RESULTS

We conducted the additional experiments on two datasets at different aspects (*i.e.*, one-shot setting, larger shot Learning t-SNE and Grad-CAM visualization of invariant causal factors.) to verify the strength of CAKE.

**One-shot Setting.** We report the comparison with baselines in the one-shot setting on `DomainNet` in Table 9 and `Office-Home` in Table 13. CAKE outperforms the SOTA methods by 1.8% and 2.3% on DomainNet (ResNet-34) and `Office-Home` (VGG-16), respectively. The performance of CAKE for one-shot learning is better than the three-shot setting, which suggests the almost best accuracy are obtained (except for P→R on `DomainNet`, A→C and C→A on `Office-Home`.). As shown later, we also employ the Resnet-34 as backbone to compare the SOTA CDAC Li et al. (2021b) in Table 15. CDAC accuracy is much lower compared with our CAKE on one&three shot setting. These observations demonstrate the robustness and generalizability of proposed CAKE once again.

**Table 9:** Accuracy(%) comparison on `DomainNet` under the settings of one-shot using Resnet34 as backbone networks.

| Method | R to C | R to P | P to C | C to S | S to P | R to S | P to R | Mean Accuracy |
|---|---|---|---|---|---|---|---|---|
| S+T | 55.6 | 60.6 | 56.8 | 50.8 | 56.0 | 46.3 | 71.8 | 56.9 |
| DANN Ganin et al. (2016) | 58.2 | 61.4 | 56.3 | 52.8 | 57.4 | 52.2 | 70.3 | 58.4 |
| MME Saito et al. (2019) | 70.0 | 67.7 | 69.0 | 56.3 | 64.8 | 61.0 | 76.1 | 66.4 |
| APE Kim & Kim (2020) | 70.4 | 70.8 | 72.9 | 56.7 | 64.5 | 63.0 | 76.6 | 67.6 |
| SSSD Yoon et al. (2022) | 73.3 | 68.9 | 73.4 | 60.8 | 68.2 | 65.1 | 79.5 | 69.9 |
| DECOTA Yang et al. (2021) | 79.1 | 74.9 | 76.9 | 65.1 | 72.0 | 69.7 | 79.6 | 73.9 |
| CDAC Li et al. (2021b) | 77.4 | 74.2 | 75.5 | 67.6 | 71.0 | 69.2 | 80.4 | 73.6 |
| PACL Li et al. (2020) | 75.3 | 74.1 | 75.3 | 65.0 | 72.1 | 68.1 | 79.7 | 72.8 |
| Baseline | 75.3 | 71.2 | 73.4 | 65.1 | 68.9 | 69.5 | 77.6 | 71.5 |
| **CAKE (Ours)** | 80.6 | 76.9 | 78.3 | 69.5 | 72.3 | 71.9 | 80.3 | 75.7 |

**Table 10:** Accuracy(%) comparison of UDA and SSDA of proposed CAKE on `DomainNet` under the settings of three-shot using Resnet34 as backbone networks.

| Method | UDA | SSDA | R to C | R to P | P to C | C to S | S to P | R to S | P to R | Mean Accuracy |
|---|---|---|---|---|---|---|---|---|---|---|
| CAKE w/o CDL | ✓ | | 77.2 | 73.0 | 74.8 | 66.0 | 69.8 | 67.6 | 79.2 | 72.5 |
| CAKE w/o ICL | | ✓ | 80.3 | 75.1 | 77.8 | 68.2 | 71.7 | 72.4 | 80.9 | 75.2 |
| CAKE w/o IPW | | ✓ | 80.1 | 74.6 | 78.1 | 69.0 | 72.0 | 72.5 | 80.8 | 75.3 |
| **CAKE (Ours)** | | ✓ | **83.3** | **77.6** | **79.1** | **72.2** | **73.0** | **74.5** | **83.2** | **77.6** |

**Table 11:** Accuracy(%) comparison of one-stage CAKE and baselines on DomainNet under the settings of 3-shot using Resnet34 as backbone networks.

| Method | R to C | R to P | P to C | C to S | S to P | R to S | P to R | Mean Accuracy |
|---|---|---|---|---|---|---|---|---|
| CDAC Li et al. (2021b) | 79.6 | 75.1 | 79.3 | 69.9 | 73.4 | 72.5 | 81.9 | 76.0 |
| PACL Li et al. (2020) | 79.0 | 77.3 | 79.4 | 70.6 | 74.6 | 71.6 | 82.4 | 76.4 |
| CAKE (w/o) $\mathcal{S}_C$ | 79.0 | 77.3 | 79.4 | 70.6 | 74.6 | 71.6 | 82.4 | 76.4 |
| **CAKE (Ours)** | **83.3** | **77.6** | **79.1** | **72.2** | **73.0** | **74.5** | **83.2** | **77.6** |

**UDA Setting.** To further evaluate the effectiveness of the proposed ICL proxy, we tested a variety of ablation models: (1) CAKE w/o CDL, (2) CAKE w/o ICL (3) CAKE w/o IPW. From Table 13, one could observe that the CAKE with UDA setting also obtains comparable results compared with SSDA models. This table also suggests the ICL and IPW are both useful to boost SSDA performance, which contributes to the accuracy of 2.3 and 2.2. These observations further demonstrate the robustness and generalizability of the proposed ICL and IWP once again.

**Table 15:** Accuracy on Office-Home for one- and three shot using ResNet-34.

| Method | One-shot | Three-shot |
|---|---|---|
| CDAC Li et al. (2021b) | 70.6 | 74.2 |
| **CAKE (Ours)** | **72.8** | **75.4** |

**Analysis of Class Imbalance.** Table 12 summarizes the comparison of generated pseudo-labels of CAKE and CAKE (w/o IWP) on `DomainNet`. As reported in Table 12, under the imbalanced labeled and unlabeled data, Our CAKE can generate more pseudo-labels with higher accuracy, both in tail class data and head class data. In contrast, CAKE w/o IWP generates few correct pseudo-labels, especially for the tail class data. These results empirically verified the robustness of the debiasing mechanism that can generate a more accurate and balanced result. Such mechanism provides a reliable solution that guarantees the mutual data knowledge exchanging from source and target aspects.

**Table 16:** Results on `DomainNet` (R → C) at *10, 20, 50*-shot setting, using ResNet-34.

| Method | 10 | 20 | 50 | Mean |
|---|---|---|---|---|
| S+T | 69.1 | 72.4 | 77.5 | 73.0 |
| DANN | 66.2 | 68.0 | 71.1 | 68.4 |
| ENT | 77.9 | 80.0 | 83.0 | 80.3 |
| MME | 77.0 | 78.5 | 80.9 | 78.8 |
| **CAKE (Ours)** | **83.3** | **84.0** | **86.2** | **84.5** |

**Larger Shot Learning**. We provide *10,20,50*-shot SSDA results on `DomainNet` (R → C) in Table 16. We randomly select and add additional samples per class from the target domain to the target labeled pool. The implementation details are the same as those of *1,3*-shot. From this table, CAKE's performance improved along with more shots and can outperform baselines from 10- to 50-shot settings, which maintains remarkable results consistently.

**IRM *vs*. ICL**. As pointed out in Sec., IRM is is not sufficient to ensure reduced discrepancy across domains. To validate our point, we report the experimental results of IRM *vs.* ICL (Table 14) and further analyses to shed light on the point. According to Table 14, ICL-based CAKE outperform IRM-based approaches by a large margin of mean accuracy (`DomainNet`:IRM +22.3% and LIRR:20.9%, `OfficeHome`, IRM: +10.3% and LIRR +7.5%). IRM asserts that it is critical to minimize the

**Table 12:** Comparison of generated pseudo-labels of CAKE and CAKE (w/o IWP) on `DomainNet` (R to C).

| | Head Class Data | PLs (Head Class) | Correct Pls (Head Class) | Tail Class Data | PLs (Head Class) | Correct Pls (Head Class) |
|---|---|---|---|---|---|---|
| CAKE (w/o IWP) | 15326 | 13512 | 12414 | 2999 | 2424 | 2264 |
| **CAKE** | **15326** | **13912** | **12866** | **2999** | **2493** | **2398** |

**Table 13:** Accuracy on `Office-Home` (%) for one-shot setting with 4 domains, using VGG-16.

| Method | R to C | R to P | R to A | P to R | P to C | P to A | A to P | A to C | A to R | C to R | C to A | C to P | MEAN |
|---|---|---|---|---|---|---|---|---|---|---|---|---|---|
| S+T | 39.5 | 75.3 | 61.2 | 71.6 | 37.0 | 52.0 | 63.6 | 37.5 | 69.5 | 64.5 | 51.4 | 65.9 | 57.4 |
| DANN Ganin et al. (2016) | 52.0 | 75.7 | 62.7 | 72.7 | 45.9 | 51.3 | 64.3 | 44.4 | 68.9 | 64.2 | 52.3 | 65.3 | 60.0 |
| MME Saito et al. (2019) | 49.1 | 78.7 | 65.1 | 74.4 | 46.2 | 56.0 | 68.6 | 45.8 | 72.2 | 68.0 | 57.5 | 71.3 | 62.7 |
| DECOTA Yang et al. (2021) | 47.2 | 80.3 | 64.6 | 75.5 | 47.2 | 56.6 | 71.1 | 42.5 | 73.1 | 71.0 | 57.8 | 72.9 | 63.3 |
| ELP Inoue et al. (2018) | 49.2 | 79.7 | 65.5 | 75.3 | 46.7 | 56.3 | 69.0 | 46.1 | 72.4 | 68.2 | 67.4 | 71.6 | 63.1 |
| Baseline | 47.8 | 79.7 | 64.8 | 75.1 | 47.5 | 56.3 | 70.9 | 42.3 | 73.4 | 71.3 | 57.3 | 72.4 | 63.2 |
| **CAKE (Ours)** | **50.2** | **82.4** | **65.9** | **76.4** | **48.6** | **58.1** | **73.7** | **45.9** | **75.0** | **74.4** | **61.5** | **74.6** | **65.6** |

discrepancy between conditional distributions over the features. However, IRM is not sufficient to ensure reduced feature discrepancy across domains and it makes the model rely more on spurious correlations (style → label). Instead, ICL not only considers the invariant risk, but also models the invariant concept by eliminating the confoundin effect of spurious correlations, which further acknowledges the importance of ICL.

**t-SNE and Grad-CAM visualization of ICFs.** To further assess the impact of invariant concept learning, we randomly select two class ( class A :"helicopter", class B : "camel" in `DomainNet` dataset, R → C scenario) samples with their invariant causal factors (ICFs) and use t-SNE to visualize their embeddings in 2D space. Overall, as shown in Figure 8 CAKE clusters instances with their ICFs are apparently classified into two classes. The ICFs preserve the invariant class-wise concept semantic with different styles that helps the SSDA model distinguish different classes. What's more, we visualize four examples with ICFs by Grad-CAM Selvaraju et al. (2017) to further examine the invariant concept learning. Figure 6 shows CAKE learns the invariant concept and attends to the similar image pixels in these samples ( same concept with different styles) , *e.g.*, foreground object shape semantics. For instance, in the "Castle" example, the styles of the four ICFs are drastically different from the original "Castle" image. Nevertheless, benefitting from the causal interventions, our CAKE can distinguish the invariant "concept" features across domains and ignores the changing of the "style", which yields improved generalization guarantees.

**Investigation for the Multi-object Scenario.** Figure 9 systematically shows the robustness of our proposed invariant concept learning when there are multiple objects in an image. For instance, in the first case (Real → Clipart) with a dog and a cat, our CAKE can attend to the most influential pixels on one concept when the classification result is "dog/cat". However, CAKE (w/o ICL) focuses on the two animals, even if one of them is the obviously extraneous factor. The truth is that the irrelevant animal plays a critical context in the prediction process for the trained classifier CAKE (w/o ICL). In other words, this extraneous animal is a part of the invariant context about the key concept, which may downplay the key concept features simply. In contrast, our CAKE is trained from a complex and changeable context with the invariant concept, so it can attend to the appropriate and unique concept even if there are two objects in the image. This provides a more reliable explanation of the proposed assumption once again.

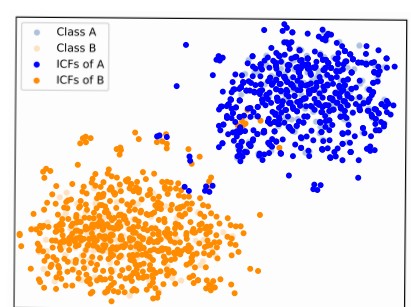

Figure 8: t-SNE visualization of two class samples (`DomainNet`, R → C ) with their ICFs.

**Table 14:** Accuracy (%) comparison (higher means better) of IRM method and CAKE on `OfficeHome` and `DomainNet` datasets.

| Method | R to C | S to R | C to S | Mean Accuracy | A to R | R to P | P to C | Mean Accuracy |
|---|---|---|---|---|---|---|---|---|
| IRM Arjovsky et al. (2019) | 60.7 | 67.8 | 52.3 | 60.27 | 74.7 | 79.7 | 59.0 | 71.3 |
| LIRR Li et al. (2021a) | 62.7 | 69.4 | 54.1 | 62.1 | 76.1 | 83.6 | 62.6 | 74.1 |
| Baseline | 81.5 | 79.6 | 70.4 | 77.1 | 79.7 | 85.6 | 65.6 | 77.0 |
| **CAKE** | **85.9** | **83.6** | **79.6** | **83.0** | **84.5** | **88.3** | **72.2** | **81.6** |

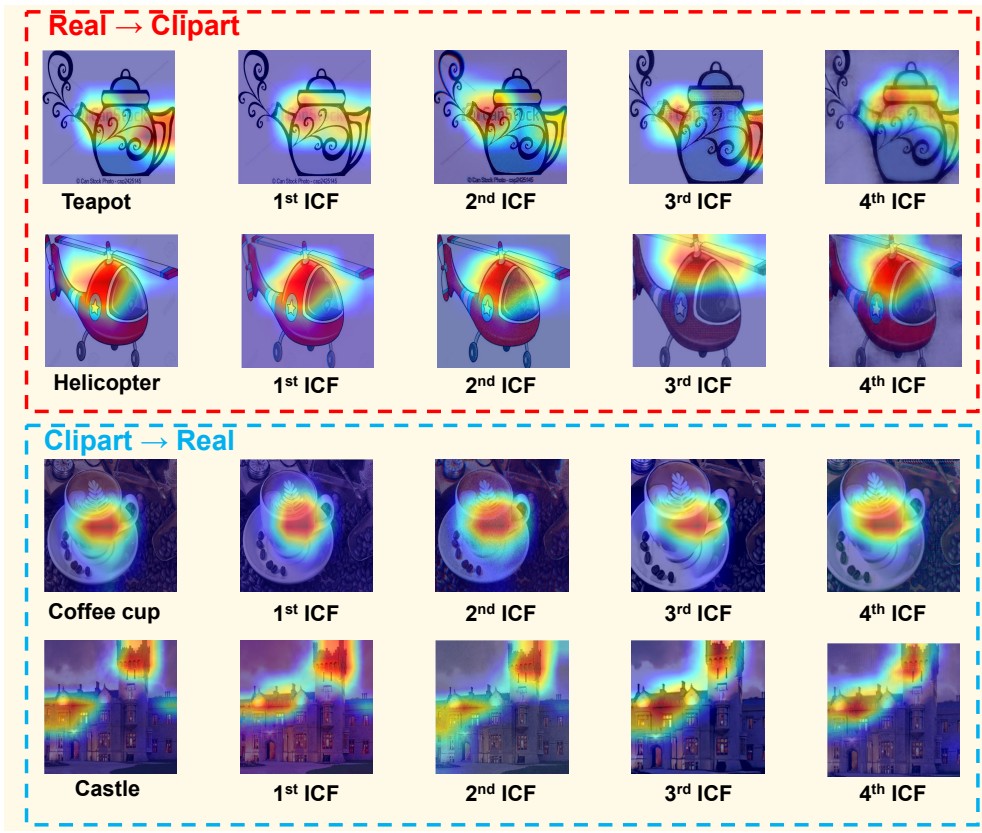

Figure 6: Class Activation Maps (CAMs) of four cases which correctly predicted by CAKE on `DomainNet` dataset (R → C and C → R scenes).

## 11 LIMITATIONS

One limitation of the proposed CAKE is the training time of invariant causal factor generation, *i.e.*, CAKE training. the training duration of cross-domain ICFG, which is responsible for generating the cross-domain style transfer samples, is long in comparison to As shown in Table 17, generating these ICFs of cross-domain style changing samples requires around 12 days, mainly due to the large-scale of the benchmark dataset of `DomainNet`. Although unfavorable, this paper focuses on the importance of the causal inference for the SSDA task. We believe that introducing the causal theoretical view into SSDA can provide new insights to the domain adaptation community. Moreover, we found that the ICFs of cross-domain style samples both preserved the invariant concept shape with different styles. We believe that an alternative way is only to use the simple image augmentation methods (*e.g.*, image transformation of brightness, temperature and sharpness) to generate ICFs to measure the deconfounded effect. As shown in Table. 11, compared with SOTA methods CDAC Li et al. (2021b), PACL Li et al. (2020) and our CAKE, obtained a comparable results that further verifies the validity of our method.

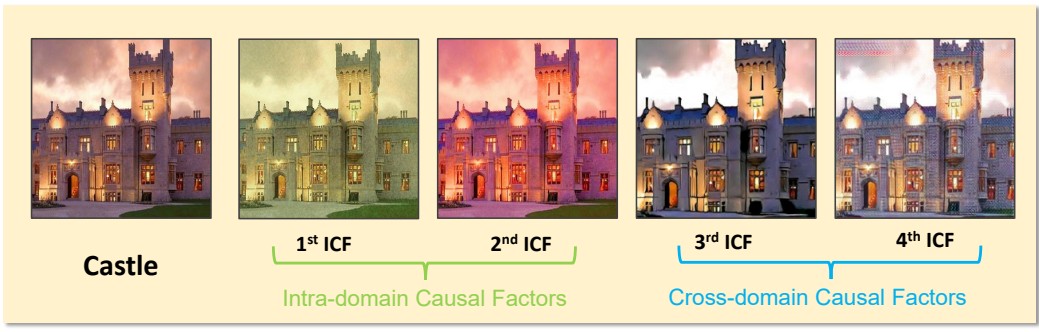

Figure 7: An example of generated intra-domain ICFs and cross-domain ICFs.

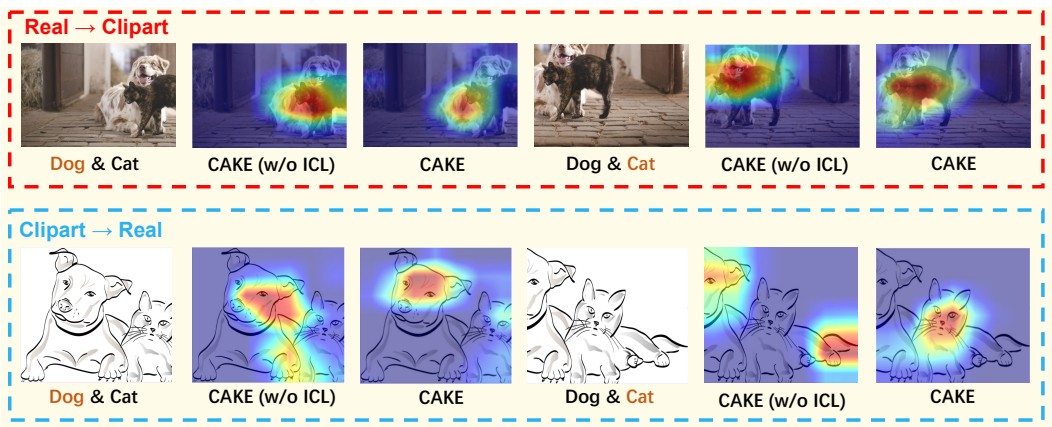

Figure 9: Class Activation Maps (CAMs) of multi-object cases which predicted by CAKE (w/o ICL) and CAKE (R → C and C → R scenes).

Table 17: Space-time complexity of CAKE

|  | DomainNet | Office-Home |
|---|---|---|
| Cross-domain ICFG Training Time | 12 days | 3 days |
| GPU Memory (Cross-domain ICFG) | Around 49GB | Around 49GB |
| CAKE Training | Around 8 hours | Around 2 hours |
| GPU Memory (CDL) | Around 81MB | Around 81MB |

