# OpenReview forum: "CAKE: CAusal and collaborative proxy-tasKs lEarning for Semi-Supervised Domain Adaptation"
_ICLR.cc/2023/Conference — Submitted to ICLR 2023_

### Official Review · Reviewer_HFEB · 2022-10-25

**Confidence:** 4
**Correctness:** 2
**Technical Novelty And Significance:** 3
**Empirical Novelty And Significance:** 3
**Recommendation:** 5

**Clarity, Quality, Novelty And Reproducibility:**

The writing quality and symbol system need to be further improved.
The originality is fine.
The clarity is not good enough.

**Strength And Weaknesses:**

Pros:
* This paper understands the semi-supervised DA from the perspective of causality, which is novel and significant.

Cons:
* This paper always tells the readers what they have done, but not their reasons. For example, in Intro, they adopt the IPW to develop the equitable pseudo-label propagator, and claim that the negative impact caused by label distribution shift can be mitigated. I think there exists a obvious gap between this technique and the label distribution shift. This type of gaps should be further clarified in the revised version.

* Some previous works (e.g., [1] [2]) have studied the DA problem through the lens of causality. However, I do not find the necessary discussions about the similarities and differences between CAKE and previous works.

* They use the CycleGAN to realize the cross-domain causal generation. However, the related analysis about why the images generated by CycleGAN are under causal interventions is missing. Actually, they use the CycleGAN to implicitly make the disentanglement and generation, and they do not learn the generative factors explicitly. Thus, clarifying this mechanism is very important.


Questions:
* Intuitively, I cannot distinguish the cross-domain style variables and intra-domain style variables in an image. If possible, please give a specific example to explain these two types of variables.

* They claim that the two types of style variables serve as the confounders. Based on the definition of confounder, it is the common causes of $x$ and $y$. However, in Figure 2, $S_I$ and $S_C$ are only the cause of $x$. Besides, they state that there exists spurious correlation between $S_I$, $S_C$ and $C$ for several times, but I cannot find this in Figure 2. I am still confused about this issue.

* Some typos. For example, in Eq.1 and Eq. 2, $P(Y|do(x),D=D_T)$ and $P(Y|do(x);D=D_T)$ appear simultaneously.

* In Theorem 1, I think $D=D_T$ indicates that the value of $D$ has been fixed. Why does this term, $\sum_{D\in\{D_S,D_T\}}$ , appear in the decomposition of $P(Y|do(x),D=D_T)$?

[1] Kun Zhang et al. Domain adaptation under target and conditional shift. ICML, 2013.

[2] Mingming Gong et al. Domain Adaptation with Conditional Transferable Components. ICML, 2016.

**Summary Of The Paper:**

This paper proposes a causal framework to formalize SSDA, and theoretically explain what characteristics should a robust domain adaptation model have. They also discuss the maximal training data utilization and present a collaboratively debiasing learning framework to make use of the training data.

**Summary Of The Review:**

Please refer to Strength And Weaknesses.

---

### Official Review · Reviewer_dEun · 2022-10-25

**Confidence:** 4
**Correctness:** 3
**Technical Novelty And Significance:** 3
**Empirical Novelty And Significance:** 3
**Recommendation:** 6

**Clarity, Quality, Novelty And Reproducibility:**

The novelty is well explored, yet the paper is not presented clearly, especially for the detail of the method.

**Strength And Weaknesses:**

Strength:
- The causal view of SSL DA sounds good.
- The proposed method is novel.

Weaknesses:
- Some typos in the paper.
- The ICT module is in an unsupervised manner, which is not very matched to the semi-supervised setting.
- The descriptions of the proposed method are not very clear. More details should be added in the supplementary.

**Summary Of The Paper:**

Focusing on semi-supervised domain adaptation, this paper considered two key points: robust domain adaptation learning and maximal cross-domain data utilization. Based on this,  a collaborative debiasing learning framework is proposed, which utilizes two complementary semi-supervised learning classifiers to mutually exchange their unbiased knowledge.

**Summary Of The Review:**

The general idea of this paper is easy to understand and some theoretic and experimental analyses could support their claims. However, whether is it really applicable or acceptable for research is hard to estimate.

---

### Official Review · Reviewer_sxkD · 2022-10-26

**Confidence:** 5
**Correctness:** 2
**Technical Novelty And Significance:** 2
**Empirical Novelty And Significance:** 1
**Recommendation:** 3

**Clarity, Quality, Novelty And Reproducibility:**

The presentation is clear.
There is a link to the code.
The novelty is limited (see the weakness above).

**Strength And Weaknesses:**

## Strength
1. The proposed method is technically sound.
2. Experiments verify the effectiveness of the proposed method.
3. The paper is well written. It presents the proposed method in an appealing and professional way with nice figures, formulas, proofs, and proper  language.


## Weakness
1. The novelty is limited. While the paper tactically presents the proposed method in an appealing way, by linking to the causality learning theory and using propositions, theorems, and proofs, the techniques in essence are quite common in this field. The so-called invariant causal factor generator is just using a GAN model to transfer labeled source image to the style of the target domain for the cross-domain causal factor, and for intra-domain causal factor using stochastic data augmentation which was used by several existing SSDA methods [1,2,3]. The collaboratively debiasing technique, i.e., using one-classier's output to supervised another classifier was also proposed before to address different problems, even for domain adaptation [4] (which is the only one I can recall at this moment, but there should be more). Linking DA to causality learning theory looks new to this problem, but it more like reiterating a well-acknowledged view in a different way that a domain can be decoupled into a domain-shared factors and domain-invariant factors. Besides, after using almost one-page trying to link DA to the causality theory, the paper concludes with the remarks that "it is non-trivial to personally determine ..." and "we employ a compromise solution", which make it doubtful for the motivation of introducing the causality stuff. In addition, SSDA is highly practical problem; it extends from UDA by adding some label supervision from the target domain to boost generalization performance. Personally I do not think many of the theorems and proofs presented in this paper are of big meaning for this practical problem. At last, the meaning of SSDA as an individual research problem is how to utilize the few labeled target samples, otherwise, SSDA can be simply reduced to a speicial case of UDA with an enlarged labeled sample set. The only technique in the proposed method that is specifically for SSDA is using two semi-supervised learners to do cross-supervision; using semi-supervised learner to learn from a mixture of labeled and unlabeled data brings little insights to help address the SSDA problem.

2. The performance is not convincing. The authors claim they "**extensively** evaluate their method" and "**significantly** outperforms SOTA methods", which I do not agree. First of all, previous SSDA methods [1,2,3 and other cited methods] usually conduct extensive experiments to verify the effectiveness and advantages over existing methods. The evaluation in this paper is far less than the previous methods, in terms of both datasets employed and backbone types. Second, previous SSDA methods usually train in one-step and can then can be deployed. The proposed method requires one extra step to train an external GAN model to generate intermediate data. The training process is more complicated, which might be ok if the results are significantly better. But as can be seen in the tables that the proposed method is only slightly better, which makes it questionable for the practical value. Last, there are other published works (e.g., [1,2]) which report better results on the employed benchmark than what are shown in the table; these works are not cited, discussed and        compared.


- [1] Cross-domain adaptive clustering for semi-supervised domain adaptation, CVPR'21
- [2]  Semi-Supervised Domain Adaptation with Prototypical Alignment and Consistency Learning, ICCV'21.
- [3]  Clda: Contrastive learning for semi-supervised domain adaptation, NeurIPS'21.
- [4] CDTrans: Cross-domain Transformer for Unsupervised Domain Adaptation, ICLR'22

**Summary Of The Paper:**

This paper addresses the SSDA problem from the perspective of causality learning. It involves two step for learning, one using GAN and data augmentation to generate intermediate data and second using two semi-supervised learner to do cross-supervision for the aim of debiasing. The effectiveness of the proposed method is demonstrated on two datasets.

**Summary Of The Review:**

This paper wraps some common techniques in some professional, appealing and smart way. The technical contribution is not enough and the evaluation is not sufficient and convincing.  I would vote for a clear rejection.

---

### Decision · Program_Chairs · 2023-01-20

**Decision:**

Reject

**Justification For Why Not Higher Score:**

the motivation of connection to causality is not convincing

limited technical novelty

**Justification For Why Not Lower Score:**

n/a

**Metareview: Summary, Strengths And Weaknesses:**

This paper studies the semi-supervised domain adaptation problem from a causal perspective. From a causal perspective, the authors propose a robust DA model to distinguish the invariant features by generating concept-invariant samples through causal intervention. In addition, this paper proposes a collaboratively debiasing framework that uses semi-supervised learning to mutually exchange their unbiased knowledge, which helps better use of source and target data.

There are several crucial concerns raised by the reviewers. One is the lack of motivation to connect the problem to causality. The causal perspective proposed in this paper is not natural and not sensible. Second, there is limited technical contribution. Most of the methods used in this paper exist in the literature, and the motivation to combine them is not strong. Finally, the proposed method does not significantly outperform existing ones. More experiments need to be done by comparing with sota methods. Given these concerns, I would recommend rejection of this paper.